# Polyureas Versatile Polymers for New Academic and Technological Applications

**DOI:** 10.3390/polym13244393

**Published:** 2021-12-15

**Authors:** Jeferson Santos Santana, Elisangela Silvana Cardoso, Eduardo Rezende Triboni, Mário José Politi

**Affiliations:** 1Centro Universitário das Faculdades Metropolitanas Unidas, FMU, São Paulo 01503-001, Brazil; jefreys2005@gmail.com (J.S.S.); acdc.elis@gmail.com (E.S.C.); 2Laboratório de Nanotecnologia e Engenharia, Escola de Engenharia de Lorena, EEL, Universidade de São Paulo, USP, São Paulo 12602-810, Brazil; tribonier@usp.br; 3Departmento de Bioquímica, Instituto de Química, Universidade de São Paulo, USP, São Paulo 05508-000, Brazil

**Keywords:** polyureas, urea polymers, symptons

## Abstract

Polyureas (PURs) are a competitive polymer to their analogs, polyurethanes (PUs). Whereas PUs’ main functional group is carbamate (urethane), PURs contain urea. In this revision, a comprehensive overview of PUR properties, from synthesis to technical applications, is displayed. Preparative routes that can be used to obtain PURs using diisocianates or harmless reagents such as CO_2_ and NH_3_ are explained, and aterials, urea monomers and PURs are discussed; PUR copolymers are included in this discussion as well. Bulk to soft components of PUR, as well as porous materials and meso, micro or nanomaterials are evaluated. Topics of this paper include the general properties of aliphatic and aromatic PUR, followed by practical synthetic pathways, catalyst uses, aggregation, sol–gel formation and mechanical aspects.

## 1. Introduction

Polyureas (PURs) are a group of polymers which contain the urea group as the main feature of their monomers, while polyurethanes (PUs) present carbamates as their functional group. Polyureas have applications in the fields of construction, new materials and several others, from health to technical fields [1].

Initial studies of polyurethanes (PUs) were carried out by Otto Bayer in 1947 [2]. These structures were compared with a series of aliphatic polyamides. Nowadays, polyureas (PURs) and polyurethanes (PUs) are used in a broad range of applications [3]. Their broad sets of properties, namely high durability and strong resistance to atmospheric, chemical, and biological factors, mean these materials have a broad range of applications [1].

The requirements for environment- and health-safe polymers are: (a) the products must undergo facile hydrolysis to create small compounds, (b) degradation products must be nontoxic, and (c) they must be biocompatible. The most commonly used biodegradable polymers include polyesters, polyamides, polyethers, PURs and PUs. PURs have not been widely studied as biodegradable polymers, but one biodegradable PUR containing trehalose residues in the main chain was synthesized by Kurita et al. [4]; PURs based on L-lysine have also been studied [5].

Polyureas for use in technical applications were derived from the reaction of an isocyanate component and a synthetic resin blend component with a “free” amine group. Polyureas can exhibit a wide range of overall properties based on the volume fraction of its components and the mixing procedure [6,7].

There are many possible uses for PURs, and they are also the best choice of material for certain specific applications. Applications for PURs include coatings for truck bed liners as well as coatings of pipelines due to their high durability and resistance to waterproofness [8]. There is also wide range of cushioning/impact applications, especially in athletic footwear and protective body/headgear systems [9], such as greases forming a thicker low-speed film [10] and porous adsorbents such as those obtained with polyamine—modified PUR used for the batch adsorption of Cu^2+^, Ni^2+^ and Pb^2+^. Porous polymer materials have the advantages of low densities, large specific surface areas, adjustable pore sizes and easy functionalization by a variety of groups, including amino, carboxyl, hydroxyl and thiols and so on, which aid their use in applications [11].

Amirkhizi et al. [12] prepared PURs to be used as a protective layer to increase the survivability of structures under extreme conditions such as impact, blasts, ballistic penetration, etc. [12,13].

In medical applications, Cusco et al. 2016 [14] recently developed a new drug delivery system for cancer chemotherapy based on PUR/PU nanocapsules that might protect the medicine from premature activation and specifically release it in tumor cells [14,15].

Lin (2016) proved that a porous PUR monolith is an effective new type of adsorbent (which has highly porous three-dimensional scaffolds with a well-interconnected mesopores and macroporous structure) for water purification that can be prepared by a reaction between only toluene diisocyanate monomer and water vapor (Figure 1). The monolith has a continuous interconnected network structure. From the SEM image at 5000× amplification (Figure 1), it is clear to see that the monolith has a skeletal structure, which is composed of fiber-like clusters [16].

Currently, two important PUR commercial niches are the automotive and construction industries, with familiar finished products such as bumpers, fascias, waterproofing linings, thermal insulation materials, industrial flooring and sports facilities [17]. PURs are often used as coatings for large surface area projects, such as secondary containments, manhole and tunnel coatings, tank liners and truck bed liners [18].

This revision presents a synopsis regarding PURs in relation to their main applications, synthetic aspects, synergy with catalysts and main properties, such as mechanical properties, supramolecular association and gel formation.

## 2. Polyureas

PURs may be classified as hetero chain macromolecules which contain urea groups in their chain. Although the chemistry and technology of PURs are of recent origin, the chemistry of urea and urea derivatives dates back over 100 years [2]. Linear PURs are in general thermoplastic materials made via the condensation of isocianates with aliphatic or aromatic amines. PUR made with aliphatic amines exhibit a difference of 50 to 100 °C between melting points and the beginning of decomposition. Accordingly, they are used for castings. On the other hand, PURs containing aromatic structures melt near their decomposition temperatures [19].

Polymers which only have urea moiety for interchain interaction have the intrinsic ability to form multiple donor–acceptor bonds based on urea H bonding linkage; thus, linear PUR H-bonds stabilize intermolecular interactions. The resulting polymer contains highly crystalline hard segments and urea–urea interactions, conferring the polymers with low solubility in most common solvents and sparse solubility in aprotic dipolar solvents, such as dimethylformamide (DMF), dimethylacetamide (DMAc) or N -methyl-pyrrolidone (NMP) [20].

The properties of PURs can be attributed to their chemical composition and the micro-phase structures. The interactions among the chains via hydrogen bonding contribute a lot to the aggregation of hard segments into hard domains [21]. Hydrogen bonds in PUR make it more flexible than materials such as epoxy or PU. When stress is applied to the cured material, the hydrogen bonds break, while the covalent polymer bonds remain intact. When the stress is removed, the hydrogen bonds reform to prevent damages [22].

Solvent choice remains critical for maintaining a homogeneous reaction and avoiding viscosity increases upon PUR formation, a critical problem for large-scale PUR production. For example, studying non-segmented poly(dimethyl siloxane) (PDMS)—containing PU from a simple addition of co-monomers, a noticeable change in viscosity as molecular weight increases was noticed [23]. Segmented PUR employs a monomeric diamine chain extender and produces PUR with distinct properties in comparison to their non-segmented analogs. Segmented PDMS-PUR synthesis traditionally requires binary solvent mixtures [23], which are usually applied to synthetic elastomers.

The hard domains of PURs are reversible physical cross-links, which play a critical role in their physical properties. Despite the reversible physical cross-links in PURs, chemical cross-linking agents can be added to improve basic structural frameworks for PURs, leading to the enhancement of the mechanical properties and resistance to solvents [21].

Isocyanates can be synthesized in many ways (Figure 2). The Curtius, Hoffman, and Lossen rearrangements can involve nitrene as an intermediate but are not useful for large-scale operations (Figure 2). The use of azides, on the other hand, is hazardous, and the utility of the Hoffman and Lossen rearrangements is limited to the preparation of aliphatic isocyanates. Alternatively, tertiarybutyl hypochloride could be used for nonaqueous Hoffman rearrangements, but the costs of this option are impracticable [24].

The most common method of producing isocyanates is “the phosgene route” (Figure 3). This technique consists of a reaction between an amine and phosgene [25]. Although harmful, the use of phosgene and phosgene substitutes is still the traditional method for the formation of symmetrical PURs in the industry. In the case of unsymmetrical PURs, the efficiency is joined with the competitive formation of the symmetrical products. In recent years, toxic and unstable reagents, such as phosgene and isolated isocyanides, ruled non safe in many policies, shall be substituted for cleaner and inherently safer alternatives [26].

A very attractive alternative is the replacement of phosgene by carbon dioxide. The mechanism of this transformation passes through iminophosphoranes, which can react with CO_2_ to generate isocyanates. This reaction is compatible with a large number of groups, and therefore, has various synthetic uses that and can be exploited for the preparation of heterocyclic compounds. Isocyanate derivatives have been obtained by this approach in good yields. However, to obtain high-purity products, it is necessary to avoid the traditional triphenylphosphine [28].

## 3. Synthesis of Polyureas

Polyurea chemistry is a relatively new synthetic process but is similar to the one used for the synthesis of polyurethanes. For PUs, the catalyst used determines the properties of the polymers which are manufactured by a polyaddition reaction between di- or poly-isocyanates and two or multi-functional polyols [29]. Due to the high nucleophilicity of amines over that of hydroxyl groups (alcohols), PURs do not require a catalyst as is required for PUs [30].

As presented by Molinos [31], the reaction of primary amines and CO_2_ to form the respective ammonium carboxylate facilitates the subsequent reaction to yield isocyanates. As the conversion of CO_2_ to isocyanates is endothermic (ΔH_R_ = +58 kJ/mol and ΔG_R_ = +51 kJ/mol) and this reaction lacks a proper catalyst, in addition to the high reactivity of the isocyanate towards several products such as urea, isocyanurate and carboiimides, this process is very unlikely (Figure 4) [31].

In general, the reactivity toward isocyanates of alcohol polymers is greater than that of monomers; hence, with increasing alcohol concentration, the relative concentration of the former is augmented, and consequently, the overall reaction process is enhanced [32]. Satchell (1988) showed that under 8 × 10^−1^ M, ethanol is only in monomeric form and the loss of reactivity is important. They considered the fact that monomeric alcohols are 1000 times less reactive than trimer forms. The authors conjectured that the lack of reactivity in the concentration range (2 × 10^−1^–2 × 10^−4^ M) is due to a loss of catalytic effect (due to a lack of nucleophile polymers) in addition to low isocyanate reactivity [33]. However, as a consequence, the addition of the isocyanate, the alcohol and amine moiety took place, resulting in bis-urea or bis-urethane compounds b and c, shown in Figure 5 [34].

The topic of stained PURs has received very little academic attention. These oligomers are amorphous in nature and they are a colored powder. They have low thermal stability. Due to the urea groups, they can be compatible with thermoplastics and can easily form colored articles, even when processed at high temperatures [35]. All the polyureas based on azo disperse dyes of the type AAB were prepared as follows, from reference [19]: “*To an ice cooled solution of azo disperse dye sample (0.01 moles) in dry tetrahydrofuran (50 mL) a solution of hexamethylene diisocyanate (0.01 mole) in 50 mL dry tetrahydrofuran was added gradually with constant stirring. The colloidal suspension which formed immediately was then stirred at room temperature for an hour*”. The resultant mixture was refluxed for 2 h, and then filtered off and dried (95% yields) (Figure 6) [19].

Synthetic or natural polymers that contain labile groups or bonds (hydrolysis or enzyme digestion) are called biodegradable polymers. Although synthetic polymers have some obvious advantages over their natural counterparts, these polymers are quite expensive and therefore cannot be used in the commercial-scale production of coated urea yet [36].

Urea is seldom used as start material in the preparation of polyurea, but Lu et al., 2016 prepared a series of novel polyurea-coated urea (PCUR) fertilizers using polyurea synthesized by the reaction of isocyanates with liquid urea as the main coating material. The granulated urea was firstly heated and changed into a liquid urea (LU). Then, LU, polymethylene polyphenyl isocyanate (PAPI) and (or) modifier were mixed uniformly to obtain the coating liquid. Then, a measured amount of the coating liquid was sprayed and the final polyurea-coated urea (PCU) was obtained. These products were denoted as PN (PUR via LU and PAPI), by including: an amino-coated PNA, ethylene glycol PNE, diethylene glycol PNG, and PNX; the others were PCUR coated. Figure 7 shows the FTIR spectra of urea, PN and PNG. Peak assignments are taken from reference [37]: (a) urea, (Figure 7c): the presence of absorption peaks at 3344, 3447 and 1157 cm^−1^ are attributed to the stretching vibration of N–H bonds. The peaks in the region of 1629–1680 cm^−1^ are attributed to the C=O stretching bond of urea. The absorption peak for urea observed at 1460 cm^−1^ represents the stretching vibration of the C–N bond (Figure 7c); (b) PN: the absorption peak observed at 3329 cm^−1^ represents the stretching vibration of the N–H bond, the peak at about 1675 cm^−1^ is due to the stretching vibration of the C=O bond (Figure 7b), the 2255 cm^−1^ peak reflects the stretching vibration from the –N=C=O bond, and finally, the absorption peaks at around 1596 and 1526 cm^−1^ might be associated with δN–H bonding. The results in spectra a and b demonstrated the occurrence of the chemical reactions between the N–H of LU and NCO groups of PAPI and the formation of PU; (c) PNG: peaks for PNG observed at 3292, 1531 and 1713 cm^−1^ correspond to the stretching vibrations of N–H, δN–H and C=O bonds, respectively (Figure 7a). In addition to these peaks, comparison with the FTIR spectra of PN, the FTIR spectra of PNG shows that new absorption peaks appearing at 2951–2899, 1418, and 1072 cm^−1^ can be assigned to νC–H, δC–H, and νC–O–C of diethylene glycol, respectively, indicating the existence of diethylene glycol in PNG (Figure 7) [37].

The general methodology for PCUR synthesis starts by heating urea particles to 50–70 °C in a rotary drum. The coating liquid is sprayed uniformly onto the surfaces of urea particles in the rotary drum and cured for approximately 5 min to produce the polyurethane coating materials [38].

The common synthetic approach to the creation of hyper-branched polymers is based on the polymerization of AB_2_ e (or AB_m_) monomers possessing complementary a and b functionalities (e.g., hydroxyl and carboxylic acid groups). Hyper-branched polyurethane (PUR) dispersions were developed in three steps, based on two-generation hyper-branched polyester, isophorone diisocyanate (IPDI) and Bis MPA (Figure 8) [39].

Later, hyper-branched poly(ether-urethane)s were reported; the methodology involved azide-type monomers. Hyper-branched polyurea and poly(urethane-urea) are also reported via relatively simpler methods, but they are fully hard segmented. More recently, segmented poly (urethane–urea) elastomers via the A2 plus B3 approach using conventionally prepared polyurethane prepolymers and commercially available triamines as A2 and B3 monomers were reported [39,40]. A potential useful poly(ether-urea) polymer that has an ether group is still unavailable. This is due to the lack of an appropriate monomer that would require: (i) a nucleophilic substitution reaction for ether formation and (ii) functional group transformation leading to a highly reactive isocyanate group or its intermediate in addition to the protection and deprotection of counter functional groups. In one study, the synthesis and characterization of the first example of hyper-branched poly(ether-urea) with aryl-aryl-ether and aryl-alky-ether connectivity and hyper-branched poly(aryl-ether-urea) copolymer was reported. Synthetic methodology involves the formation of blocked isocyanate and azide groups and does not involve a protection–deprotection strategy for a counter functional group, that is, for amine (Figure 9 and Figure 10) [41].

Fiber structures presented by polymers usually require regular chain monomer structures. Accordingly, as described by Li [42], they used 2 mol of 4,4-Diphenylmethane diisocyanate (MDI) with 1 mol of diamine to form NCO-terminated prepolymer (step 1 in Figure 11). In the second step, different mole ratios of aromatic/aliphatic diamine were used to form pure aromatic polyurea, aliphatic polyurea and their co-polymers, as shown in steps 2, 3 and 4 (Figure 11) [42].

The monomer composition of their samples is listed in Table 1 [42].

Polyurethanes and polyureas have versatile material properties. Compared with PUs, PURs can have stronger inter-molecular interactions between PU chains. Urea moieties also have more stability than carbamic groups. Polyurethanes and polyureas can be rigid, semi-rigid or flexible depending on the monomers and the polymer microstructures. Their industrial preparations are based on isocyanate chemistry, where the isocyanates are reacted with diol or diamine functional groups, respectively [43].

Locatelli et al., 2015 presented the following recipe for the preparation of polyurea nanoparticles: 4,4′-MDI, poly(PO/EO) monoamine and diamines were dissolved separately in toluene (10% wt). The 4,4′-MDI and bisaniline solutions were prepared at 100 °C and 40 °C, respectively, and then allowed to cool down to room temperature. The other solutions were prepared at room temperature. The polyether–polyurea–polyether nanoparticles (PNPs) were synthesized via a two-step reaction. First, the polyether monoamine and the diamines (poly(PO/EO)) solution was added drop by drop to the 4,4′-MDI solution with stirring. During the second step, the diamine solution was also added drop wise to the resulting solution, with stirring as well. The reactant had a mole ratio of 1: 1 between the NCO from the 4,4′-MDI and the NH_2_ from the poly(PO/EO) monoamine and the diamines. A diamine/monoamine molar NH_2_ ratio of 40 (i.e., diamine/monoamine molar ratio of 20) was used in their work [44]. Following this, a slight excess of diamine was added (0.1% mol) to ensure the complete reaction of the NCO groups. The final polyurea PNP suspensions were stored at room temperature [44].

Hirai (1999) prepared semiconductor nanoparticle–PUR composites using reverse micellar systems via in situ diisocyanate polymerization. As compared to semiconductor nanoparticles in reverse micelles and also CdS nanoparticles surface-modified with thiols, PUR and polythiourethane composites are attractive for use in photocatalysts [45].

Field-effect transistors (FETs) are simple devices composed of three contacts (source, drain and gate), a dielectric layer and a semiconducting layer. FETs essentially act as electronic valves by modulating the semiconductor channel conductance via the gate field [46]. A modified PUR with a pyridyl group was prepared for a FET device. Undoped poly(pyridylureas) are considered semiconductors (σ = 10^−9^ (Ω cm)^−1^), but after doping with I_2_, the electrical conductivity increases by several orders of magnitude (σ = 10^−7^ (Ω cm)^−1^). Poly(pyridylureas) and poly(pyridylthioureas) were obtained by reacting phosgene or thiophosgene with 2,6-diaminopyridine, using pyridine or THF as the solvent, according to the reaction below (Figure 12) [47].

A detailed recipe for this FET-like PU-Pyridyl and poly(pyridylthioureas) is found in reference [47].

Xue-Lian (2008) [42] synthesized polyurea samples in two stages, so that the sequence structure of the chain could be controlled more regularly. In the first step, 2 mol of MDI reacted with 1 mol of diamine to form NCO-terminated prepolymer, as shown in step 1 (Figure 11). In the second step, different mole ratios of aromatic/aliphatic diamine were used to form pure aromatic polyurea, aliphatic polyurea and their co-polymers, as shown in steps 2, 3 and 4 (Figure 11). The Tg of the polyurea samples were measured by DSC and analyzed by the Fox model (Equation (1)). Figure 4 shows that the Tg of pure aromatic polyurea is 132 °C, whereas pure aliphatic polyurea is 101 °C. The glass transition temperatures of co-polyurea samples decrease with the increase in 1,6-diaminohexane (1,6-HDA) content. The Tg data of co-polyurea samples are shown in Figure 13 [42].
(1)$$\frac{1}{\mathrm{Tg}}=\frac{1-W_{2}}{{\mathrm{Tg}}_{,1}}+\frac{W_{2}}{{\mathrm{Tg}}_{,2}}$$
where Tg,_1_, Tg,_2_ and Tg are the glass transition temperatures of pure aromatic W_2_polyurea, pure aliphatic polyurea and the co-polyurea, respectively. W_2_ is the weight fraction of the aliphatic polyurea component in the co-polymer. In the case of pure aromatic polyurea, the thermal degradation temperature is 290 °C, whereas the thermal degradation temperature of pure aromatic polyurea is 357 °C. On the other hand, the thermal degradation temperatures of the co-polyurea samples decrease significantly to 250–260 °C, indicating that the incorporation of the 1,6-HDA segment significantly reduced the thermal stability of co-polyureas. According to the Freeman–Carroll equation (Equation (2)), the thermal degradation activation energy of the polyurea samples can be estimated from the data of initial thermal degradation of the samples, where E_d_ is the thermal degradation activation energy, α is the weight loss fraction, R is the gas constant, and T is the absolute temperature [42].
(2)$$\frac{\Delta \ln\left(d\alpha /\mathrm{dt}\right)}{\Delta \ln\left(1-\alpha \right)}=-\frac{E_{d}}{2.303R}\times \frac{\Delta \left(1/T\right)}{\Delta \ln\left(1-\alpha \right)}+n$$

Comparative TGA of EDA-based polyurea microcapsules with metribuzin as a core is shown in Figure 14. Metribuzin is a known herbicide, the encapsulation of which would bring potential uses in health and environment fields. The compositional analysis of microcapsules in terms of percentage was carried out using TGA data and the calculation of derivative weight loss [48].

Pure metribuzinon was analyzed via TGA, and it was found that the loss in weight started at 140 °C and finished at about 280°C in a single stage (Figure 14A). The TGA thermogram of microcapsules showed weight loss in two steps, with a rise in temperature (i.e., 140–310 °C, 310–630 °C) (Figure 14B). The first substantial weight loss (approximately 58%) in microcapsules was observed at about 140–310 °C and was attributed to loss due to the hydrophobic core material along with solvent (xylene). The second stage of weight loss was due to shell wall degradation of EDA-based polyurea microcapsules and was found to be about 40% in the temperature range of 310–630 °C. The thermal stability of polyurea was extended up to 300–350 °C, which is higher than the phenol formaldehyde and urea formaldehyde shell microcapsules with thermal stabilities around 280 °C [49].

Polyureas, which have good thermal stability, chemical resistance and good mechanical properties, can be exploited as matrices for high-performance advanced composite materials, as membranes for gas separation and as coatings. A synthetic electro-active polyurea that revealed valuable electro-chromic performance with a high contrast value, moderate switching times, acceptable coloration efficiency and excellent stability was prepared by Chao’s group [50].

Electrochemical cyclic voltammetry (CV) is widely used to characterize the redox properties of electro-active polymers, as in the study by Yeh et al., 2013 [51]; their polymer was characterized by CV using a typical three-electrode electrochemical cell. Figure 15 shows the CV curve of the electro-active polyurea (EPU), a three-pair redox peak (272, 499, and 638 mV), which is different from the typical two-pair redox peaks (350 and 800 mV). Four different oxidation states are shown in Figure 15. CV measurements of EPU measured in aqueous H_2_SO_4_ (1.0 M) at a scan rate of 50 mVs^−1^ of EPU can be observed and attributed to: Leucoemeraldine base (LB), emeraldine base I (one quinoid ring in oligoaniline segment, EBI), emeraldine base II (two quinoid rings in oligoaniline segment, EBII) and pernigraniline base (PNB). For the CV curve of EPU, the first oxidation peak corresponded to the transition from LB to EBI, the second peak corresponded to the transition from EBI to EBII and the third peak corresponded to the transition from EBII to the PNB form [51].

The synthesis of polyureas usually requires highly toxic polyisocyanates, which are derived from even more toxic phosgene materials. It would be an appreciable strategy if CO_2_ could be used as a starting material to replace the toxic polyisocyanates in the production of PUs (Figure 16) [18].

However, this route has been barely investigated since the first patent in 1951, except for the investigations of Yamazaki and Rokicki in the 1970–1980s. They used stoichiometric catalysts, such as diphenyl phosphate, phosphorus chlorides or N-acylphosphoramidites, in pyridine or acetonitrile medium. Herein, a new process for the synthesis of PURs from diamines and CO_2_ without the use of any catalyst or solvent is included [18].

Shang (2012) [48] describes that in preliminary study, hexamethylenediamine (HDA) was selected as a model substrate to test this protocol. The CP/MAS^13^C NMR spectrum of the solid product, which was produced from the reaction of HDA and CO_2_ in the absence of a catalyst, showed a characterization peak at 159.7 ppm, indicating the formation of carbonyl in the urea linkage. Furthermore, the structure of the solid product was also characterized by FTIR spectroscopic analyses (Figure 17).

The peaks at 1618 and 1577 cm^−1^ are assigned to the amide I (C=O) and amide II (CO–N–H), respectively, a clear indication of the formation of the urea moiety. Interestingly, the vibration of C=O and N–H observed at 1618 and 3329 cm^−1^, respectively, suggests that the solid product was made with ordered H bonding, as indicated in Figure 17b. From these data, it could be proposed that the solid product based on HDA and CO_2_ has a PU structure with the urea linkage and is connected by the ordered H bonding, as schematically shown in Figure 18. Accordingly, the polyurea derivative (denoted as polyurea-HDA) could be successfully produced from the reaction of HDA and CO_2_ in the absence of a catalyst [48].

The use of NMR in the study of PURs is difficult due to their low solubility in most common solvents due to the presence of hydrogen bonds between their chains [52]. An alternative study displayed in Figure 19 shows the ^1^H-NMR spectra of polyurea samples. In this study, detailed assignments are presented. in aromatic polyurea, (a) a strong signal at 3.8 ppm (1) corresponds to the protons of methylene linked with a phenyl ring of MDI (methylene hydrogen, labeled 1 in the top of the spectra and 1 as well in the urea monomer), whereas the transitions at 7.0–7.6 ppm (4, 6 and 8) are associated with the phenyl ring protons of MDI and m-phenylenediamine (m-PDA) (aromatic hydrogen labeled 4, 6 and 8 in the top of the figure and 4, 6 and 8 in the monomer molecule). Further, the peaks at 8.5 and 8.7 ppm (2 and 3) correspond to the associated aromatic urea group protons (hydrogen atoms labeled 2 and 3 in the top and bottom of the figure). In the case of aliphatic polyurea (e), the resonance peaks associated with the phenyl ring of MDI are still in the range of 7.0–7.6 ppm, while those resonance peaks associated with the urea group shift to 8.4 and 8.8 ppm (9 and 10). The protons of methylene on the 1,6-diaminohexane (1,6-HDA) unit resonate at around 1.2–1.5 ppm (11 and 12). In the case of co-polyurea samples (b–d), there are three proton peaks at 8.3, 8.5 and 8.8 ppm, respectively. This is because the urea linkages in the co-polyurea are in three different modes. It can also be found that the proton resonance of methylene on the 1,6-HDA unit becomes stronger as the content of 1,6-HDA increases. Data from reference [42].

Oligomers have a special molecular structure and limited molecular mass constituted of a set of similar or different units that are repeatedly connected with each other. Sharzaehee (2020) prepared water-soluble oligomers synthesized using urea, phosphorous acid and sulfamic acid in various molar ratios in melt conditions at a maximum temperature of 150 °C, while the evolved gas was removed by a condenser [53].

Urea oligomers can be successfully prepared in a one-pot reaction. Although chromatography is necessary to separate these oligomers, this route is rapid and has advantages over the multi-step method that uses protection groups. The trimer and tetramer are the most active oligomers. As shown in Figure 20, when 3 is reacted in excess, this approach successfully yields dimers, trimers and tetramers, as well as a fraction containing pentamers through to octamers [54].

As depicted in Figure 21, the polymerization of oligomeric PDMS diamines, urea and the optional 1,3,-bis(3-aminopropyl)tetramethyldisiloxane (BATS) yielded non-segmented (poly (PDMS-co-urea)) and segmented copolymers (poly(PDMS1.7kU)x-copoly(BATSU)y). Given the absence of an appropriate solvent, these reactions were conducted above the melting point of urea (133–135 °C). In contraposition, urea decomposes above 150 °C into ammonia and isocyanic acid, which reacts with primary amines and forms the desired 1,3-dialkylurea linkages in the absence of isocyanates. Therefore, care must be taken to avoid side product formation, which in the context of linear polyureas includes urea biurets, 1,1-dialkylurea or imidazolidone cycles. This only includes the use of primary amines, a stoichiometric excess of diamine vs. urea and temperatures in excess of 200 °C [23].

Recently, electroactive polymers incorporated with aniline oligomers have attracted research attention because of their superior properties, such as good solubility, mechanical strength and the ability to form film-conjugated oligoaniline through polycondensation or oxidative coupling polymerization. These copolymers contain well-defined conjugated segments and also provide an opportunity to present a clearer understanding about the structure–property relationships and the conducting mechanism of conjugated polymers [51]. The synthetic routes for the preparation of oligoaniline, EPU is shown in Figure 22, taken from reference [51].

New pyridine-containing diisocyanates were synthesized. Substituted pyridine containing diacids (DA1–3) were synthesized using the modified Chichibabin pyridine synthesis via the reaction of 4-methylacetophenone with substituted benzaldehydes and the subsequent oxidation of resultant dimethyl compounds [55], as shown in Figure 23.

In their work, Tamami and Koohmareh [55] synthesized the diacids and then converted them to the corresponding 4-aryl2,6-bis(4-isocyanatophenyl)pyridines (DIC1−3) by Weinstock modification of Curtius rearrangement using triethylamine, ethylchloroformate and active sodium azide reagents. The intermediate diacylazides were subjected to thermal decomposition in dry benzene at reflux temperature to yield the diisocyanate monomers [55].

Attempts to synthesize new types of thermally stable polyureas such as phosphorus-containing and heterocyclic polyureas to obtain different properties have been reported [56]. This particular study is concerned with the synthesis of 4-(4-dimethylaminophenyl)-1,2,4-triazolidine-3,5-dione (DAPTD) as a new heterocyclic monomer and its polymerization reaction with commercially available diisocyanates [56].

### 3.1. Monomer Synthesis

A comprehensive route of the heterocyclic monomer DAPTD was synthesized in five steps, starting from 4- dimethylaminobenzoic acid **1**. The synthesis of acyl azide **2** was prepared by “one-pot” Weinstock modification of the Curtius’ reaction. In this example, the isolation of intermediate was not required, and the acyl azide was obtained in a comparatively purer form and in good yield. Next, the acyl azide was subjected to thermal decomposition in dry toluene (reflux temperature) to yield isocyanate **3**. Subsequently, this isocyanate was reacted with ethyl carbazate and semicarbazide **4** was obtained in a quantitative yield. The cyclization of compound **4** with sodium ethoxide gave new urazole **6** (Figure 24) [56].

The purity of monomer **6** was checked by TLC. Their 1 H-NMR spectrum was also correct. The structure of urazole **6** was also confirmed by IR, UV–vis, fluorimetric, mass spectra and elemental analysis, taken from reference [56].

### 3.2. Polymerization Reactions

Once the monomer **6** as a model compound was obtained in high yield and purity, it become attractive to develop new photoactive polyureas. Thus, hexamethylene diisocyanate (HMDI) **8**, IPDI **9** and toluene-2,4-diisocyanate (TDI) **10** were selected. The polymerization reaction of monomer **6** with these diisocyanates was performed under conventional solution polymerization techniques as well as high temperature in the presence of different catalysts PU1-PU3 (Figure 25). Following reference [56], the polymerization reaction of **6** with HMDI was carried out with two different methods. In method I, the reaction mixture was heated gradually from room temperature to 85 °C in the presence of pyridine, dibutyltin dilaurate and triethylamine. The resulting polyureas, PU1A-PU1C, have good inherent viscosity and high yield. In method II, the reaction mixture was refluxed up for 1, 3 and 6 min in DMAc [56].

The resulting polyureas, PU1D-PU1H, have high yield and good inherent viscosity in comparison with method I [56].

### 3.3. Synthesis of Monomers

As starting materials for the preparation of diisocyanates, the stable diamino dihydrochlorides of the dianhydrohexitols with all three possible configurations (d-gluco 4, lido 5, d-manno 6) were prepared according to known procedures (Scheme 1) (Figure 26) [57].

Phosgene was used to transform the diamino compounds into the diisocyanates. This phosgenation procedure was performed in two steps, combining cold and hot phosgenation conditions. Firstly, anhydrous phosgene was added in two equimolar excess to a cooled suspension of the corresponding diamino dihydrochloride 4, 5 and 6 in anhydrous toluene (cold phosgenation). The reactions were monitored using IR spectrometry. The intermediates, probably mixtures of dicarbamic acid dichloride and dicarbamic acid dichloride dihydrochloride, were refluxed to give the corresponding diisocyanate (hot phosgenation). Under these conditions, the formation of polyureas as a side reaction could be avoided. A simple filtration step with charcoal afforded the diisocyanates dianhydrohexitols with d-gluco (7), lido (8) and d-manno configuration (10) as analytically pure materials. The yields of the phosgenation were dependent on the stereochemistry of the dianhydrohexitols. In contrast to the yields which were obtained by nucleophilic substitution reactions at the hydroxy groups at the positions 2 and 5, the addition of phosgene proceeded better when the amino groups were orientated out of the molecular plane (exo configuration). Consequently, the synthesis of the l-ido (exo, exo)-configurated diisocyanate 8 gave the highest yields, the synthesis of the d-gluco (exo, endo) 7 diisocyanate gave a 64% yield, whereas the stereochemically unfavored d-manno (endo, endo) configuration 10 only gave a moderate yield, 41%. Alternatively, we used diphosgene for the formation of diisocyanates in order to avoid the application of pure phosgene. It is known that diphosgene gave high yields only when aromatic amines and activated aliphatic amines were treated. By reaction of the l-ido configurated diamino compound 5 with three molar equivalents of diphosgene in anhydrous dioxane under the same conditions used for the phosgene reaction, the corresponding crude diisocyanate 8 had to be purified by distillation and was isolated in only 5% yield. Additionally, we synthesized one dithioisocyanate derivative of the dianhydrohexitols to explore the reactivity of the sulfur-containing monomer. For the stereochemical reasons already described, the l-ido-configurated diamino compound 5 was reacted with thiophosgene to give the dithioisocyanato compound 9 in 93% yield. The reaction was monitored using IR spectrometry and no formation of poly(thiourea) was observed. Impurities were removed so that analytically pure compounds were obtained. The diisocyanate 7, 8 and 10 were treated with selected aliphatic and aromatic monomers in the polyaddition procedure described. As petrochemically derived monomers, one diol derivative (1,4-butanediol), one dithiol derivative (1,4-butanedithiol), one aliphatic diamine derivative (1,4-diaminobutane) and one aromatic diamino monomer (1,3-diaminobenzene) were used as monomers. Furthermore, as dianhydrohexitol monomers, the diol 1,4:3,6- dianhydro-d-sorbitol (1) and the diamino compound 2,5- diamino-2,5-dideoxy-1,4:3,6-dianhydro-d-sorbitol (4) were selected for polyaddition (Figure 27, Figure 28 and Figure 29) [57].

The polyaddition reaction with 1,4-butanediol and the corresponding diisocyanates 7, 8 and 10 afforded the polyurethanes 15 with l-ido, 23 with d-gluco and 26 with d-manno configuration. With 1,4-butanedithiol, the l-ido-configurated poly(thio-urethane) 11 was accessible. 1,4-Diaminobutane was reacted to give the l-ido polyurea 13. Using 1,3-diaminobenzene, polyureas of l-ido (17), d-gluco (24) and d-manno (27) configuration were prepared. The polyaddition of 7 with 1,4:3,6-dianhydrosorbitol (1) as monomer compound led to polyurethane 19 with a heterocyclic backbone completely derived from carbohydrate material. The corresponding polyureas of l-ido (21), d-gluco (25) and d-manno (28) configurations were obtained by the polyaddition of the corresponding diisocyanates 7, 8 and 10 with 2,5-diamino-2,5-dideoxy1,4:3,6-dianhydro-d-sorbitol (4). In addition to this, the polyaddition reactions with the described aliphatic and aromatic monomers were performed using the l-ido dithioisocyanate 9. The corresponding polymers containing thiourethane (16, 20), thiourea (14, 18, 22) or dithiourea linkages (12) were accessible straightforwardly. The reactivity and yields were similar to those reported for diisocyanate 8 [57].

### 3.4. Synthesis of Polymers

The novel diisocyanato monomers 7, 8 and 10 and the dithiocyanato compound 9 were subjected to polymerization. As a polymerization technique, the polyaddition procedure was performed. As polyaddition catalysts, various Lewis bases (tertiary amines, i.e., 1,4-diazabicyclo-[2.2.0]octane or triethylamine) as well as Lewis acids (especially tin organic compounds) have been used [17,18,19]. The mechanism of the catalysis is not completely understood, which means that for every polyaddition, the best suitable catalyst and solvent system has to be screened. For most polyaddition reactions, dibutyltindilaurate was used. As a solvent, *N*,*N*-dimethylacetamide (DMAc) was preferred. With other aprotic polar solvents such as *N*,*N*-dimethylformamide (DMF) or dimethyl sulfoxide (DMSO), no precipitation of the polymer was observed, but gel-formation was observed. Advantageously, following the polyaddition procedure, polymers were obtained in good to almost quantitative yields. Similarly to the stereochemical effects during the addition reaction of phosgene to the corresponding diaminodianhydrohexitols, the stereochemistry of the monomers has an influence on the polymer yield. The higher the number of exo-configurated isocyanato groups present, the higher the yield of the polyadducts was. The polyaddition reaction of the l-ido compound 8 and 9 proceeded quantitatively; the d-gluco diisocyanate 7 gave a yield of 90%, whereas the yield dropped to 80% when the d-manno diisocyanate 10 was reacted. Moreover, the poly adducts in monomer synthesis were free of any inclusions of monomers, water or other inclusions [57].

The synthesis of selected PURs was accomplished as presented in reference [58].

“Polyurea 1—The peptide TFA.H-Leu-Tyr-.Jeffamine-Tyr-LeuH.TFA (1 mmol, 1.18 g) and TEA (4 mmol, 4 g) were dissolved in 4 mL of dry DMF. To the mixture was added hexamethylene diisocyanate 0.16 mL (1 mmol, 0.18 g). After about 1 min, the viscosity raised sharply, and to avoid it more DMF (~4 mL) was added to assure mixing and stirring. The polymerization reaction was maintained at room temperature for 72 h. The reaction mixture was poured into distilled water and a white precipitate was obtained and filtered out. The precipitate was washed with chloroform and dried in a vacuum oven yielding 0.8 g (59%) of white fibrous polymer” [58].

“Polyurea 2—The peptide TFA.H-Leu-Tyr-,Jeffamine-Tyr-LeuH.TFA (1 mmol, 1.18 g) and TEA (4 mmol, 4 g) were dissolved in DMF (10 mL)and methylenedi-p-phenyl diisocyanate (pMDI) (1 mmol, 0.25 g) was added. The mixture was kept at room temperature for 24 h. To ensure the polymerization, more MDI (0.2 mmol, 0.05 g) was added to the mixture and proceeded for 48 h; the product was similar as that of Polyurea 1. The reaction yielded 1.0 g (70%) of fibrous white solid” [58].

“Polyurea 3—To a solution of monomer 2 (1.00 mmol, 0.74 g) and triethylamine (2.2 mmol, 0.31 mL) in NMP (2.7 mL), a solution of hexamethylene diisocyanate 0.16 mL (1 mmol, 0.18 g) in chloroform (2 mL) was added with the purge of argon. The reaction proceeded overnight at room temperature. The viscous brownish sample was poured into a beaker containing about 1 L of distilled water and the product was precipitated. The polymer was washed in water, methanol, and acetone, and then dried in a vacuum (Figure 29, Figure 30 and Figure 31)” [58].

The synthesis of polyureas is attractive, as it avoids the use of toxic and reactive diisocyanates, such as MDI or toluene diisocyanate (TDI) (Figure 32) [48].

Microwave irradiation could be an alternative method used to synthesize polyureas; however, the necessity of solvents with high boiling points, such as dimethylacetamide or dichlorobenzene, becomes an additional health and environmental barrier [59].

However, in a non-isocyanate route (NIR) to polyurea, the approach based on MDI-biscarbamates seems promising because no high-temperature condition is required through trans-esterification or trans-ureation. In an initial NIR process development, the trans-ureation of 4,4′-DM-MDC and 4,4′-DP-MDC was carried out by mixing the same equivalent of polyetherdiamines, such as 1,8-diamino-3,6-dioxaoctane (DADO), at 160 °C in the absence of a solvent and catalyst [17]. During the melt-polymerization, methanol or phenol was flushed out from the reaction mixture by continuous purging with a nitrogen stream under reduced pressure. Powdered Polyurea [4,4′-DM-MDC/DADO]melt/160, (P-1), were isolated after 60 min of reaction, and GPC analyses suggested that P-1 had M.W. of 6000~7000 kDa. These M.W.s indicated a low degree of polymerization that could be attributed in part to the poor mixing between high melting biscarbamates and diamines and the high viscosity of resulting polyurea. On the other hand, if melt-polymerization was carried out at temperatures higher than 200 °C to reduce the product viscosities, Polyurea [4,4′-DM-MDC/DADO] melt/200 turned dark rapidly due to degradation and oxidation, and the molecular weights also did not increase. Thus, directly mixing biscarbamates and diamines by melt-polymerization failed to yield high molecular weight polyureas [17].

In 1999, Thavonekham [49] found a facile transureation method of N-phenyl phenylcarbamates with *N*,*N*′-di-butyl amine as solvent, and showed that the urea product could be prepared in high yield. It was observed that solvents with high polarity or high solubility parameters were the most efficient, and DMSO was found to be the best solvent among those tested [49]. In this study, trans-ureations of both biscarbamates with 2-aminoethanol in DMSO were carried out at 80 °C as in the model reaction study before their new polyurea synthesis. The result confirms the high yield (94%) of 4,4′-diphenylmethanebis-[(2-hydroxyethyl)urea], (DPMHU), which could be isolated in a short reaction time from 4,4′-DP-MDC (60 min). Conversely, under an identical condition, transureation of 4,4′-DM-MDC failed to give the urea product (DPMHU), and the un-reacted 4,4′-DM-MDC could be recovered from the mixture. These results strongly suggest that 4,4′-DP-MDC is a more effective intermediate than 4,4′-DM-MDC for carrying out trans-ureation reactions to make urea derivatives. The difference in reactivity between these two biscarbamates reflects the fact that the phenoxy group of 4,4′-DP-MDC is a better leaving group than the methoxy group of 4,4′-DM-MDC when 4,4′-DP-MDC were attacked at the carbonyl center by nuclephilies. The use of polar solvents such as DMSO seems to further activate the urethane bond in 4,4′-DP-MDC for the easy replacement of phenol. At 80 °C, tetramethylene sulfone (TMS) behaves like DMSO and also performs as the solvent for making DPMHU (Figure 33, Figure 34 and Figure 35) [17].

In the polymerization of dimethyl 1,4-butylenedicarbamate (BU2) and dicarbamates and diamino-terminated poly(propylene glycol) (PPGda), performed in the study presented in reference [60], the concentration of the starting materials applied was ~80 wt%. The polymerization proceeded smoothly in a short time (4 h). From the list of solvents that dissolve BU2 (Figure 37, Table 1 from reference [60]), DEGDE is the best choice since it is non-toxic and has a sufficiently high boiling point (180–190 °C) to allow the synthesis of PUs at elevated temperatures and ambient pressure.

Dimethyl ((carbonylbis(azanediyl))bis(1,4-butylene)dicarbamate (BU3) dissolves in DMAc, which is commonly used in the PU industry, although the solubility is not high. Therefore, the concentration of the polymerizations involving BU3 in DMAc can only be as high as 20 wt%. The polymerization of dimethyl (6,13-dioxo-5,7,12,14-tetraaza1,18-octadecylene)dicarbamate (BU4) could only be performed in NMP at a concentration of 15 wt% (Figure 36). PURs with HSs containing four urea groups have flow temperatures (Tfl) above 200 8C, which is higher than the degradation temperature of PURs. Therefore, in their study, the authors [60] did not focus on the investigation of the properties of the polymer based on BU4 and PPGda, as the material cannot be melted to form films or testing bars. The results included in Figure 37 (Table 1 from reference [60]) indicate that all the targeted polycondensation reactions proceeded and that the average molecular weight (Mns) of the resulting PURs is about 30 kg·mol^−1^. However, some of the materials have a PDI lower than 1.70 (the theoretical value for linear step-growth polymers is 2), which might be explained by the removal of part of the low molecular weight polymer during the purification (by precipitation) which is required to remove the TBD [60].

The result of Figure 37 shows that all the target polycondensation reactions were carried out, and the number average molecular weight (Mns) of the obtained PUR is about 30 kg·mol^−1^. However, the PDI of some materials is lower than 1.70 (the theoretical value of linearly increasing polymers is 2), which may be due to the removal of some low molecular weight polymers during the purification process (through precipitation) and the deletion of TBD [60].

## 4. Catalysis Applications

A relatively recent study on catalysis applications is presented in the study of the copper-catalyzed amidation of aryl halides with urea. A preliminary screening of ligands showed that both 1,2-diaminocyclohexane and *N*,*N*_0_-dimethylethylenediamine afforded products in the coupling reactions, with the latter giving low yields. Efforts to use other ligands such as ethylenediamine and 1,2-ethanediol were not successful. 1,2-diaminocyclohexane was chosen as the preferred ligand for the coupling reactions. The reaction of iodobenzene with urea in the presence of the CuI, 1,2-diaminocyclohexane catalyst system afforded the biphenyl urea 3 with a good yield (Figure 38) [61].

An effective route for the syntheses of polyurea derivatives from diamines and CO_2_ in the absence of any catalyst, which are then reacted with dialkyl carbonates to synthesize N-substituted dicarbamates over a MgO–ZnO catalyst, is shown in Figure 39.

Since the polyurea derivatives and dialkyl carbonates could be successfully synthesized from CO_2_, this process not only improves the functional group efficiency of the reagents, but also optimizes the utilization of CO_2_. In work by Shang et al. [48] the reactions were carried out in a 90 mL stainless steel autoclave with a glass tube inside and with a magnetic stirrer. Specifically, 5 mmol PUR, 10–75 mmol dibutyl carbonate (DBC) and 3–15 wt% catalyst (based on the mass of charged polyurea) were charged successively into the autoclave. The reaction proceeded at 180–220 °C for 3–24 h (using a nitrogen atmosphere) and the vessel was cooled to room temperature. The catalyst and the unconverted PUR were separated by filtration and then thoroughly dried in vacuum. The resulting solids were weighed and the mass of catalyst was subtracted to determine the conversion of the PUR [48].

Some kinetic parameters were evaluated by Suresh’s group [62] on the study of PU formation via interfacial poly condensation represented by the condensation of hexamethylene-1,6-diamine (HDMA) present in the aqueous phase, and hexamethylene-1,6-diisocyanate (HDMI) in the organic phase. This study presented good PUR microcapsules, shown in Figure 40, and presents details and methodology for interfacial PUR formation [62].

## 5. Sol-Gel and Supramolecular Aspects Applied in Polyureas

Over the past few decades, there has been a huge amount of research focused on supramolecular gels, in which organic small molecules self-assemble into a wide variety of entangled nanostructures to form a 3D network. During the self-assembly, hydrogen bonding, π−π stacking, electrostatic interactions, dipole–dipole interactions, van der Waals interactions, and metal-to-ligand coordination between the gelators play important roles. Meanwhile, gelator–solvent interactions are able to subtly regulate these interactions and thus make the gel system a good platform for the fabrication of well-defined stimuli-responsive or smart soft materials [24,63].

The gelator molecules generally bear aromatic rings, long alkyl chains sugar, OH, COOH amide groups and urea moieties, in which hydrogen bonding sites are plenty. There has been burgeoning literature describing the types of gelators according to the main driving forces for forming supramolecular gels and their corresponding assembly architectures. Generally speaking, the structural factors of the gelators determines the arrangement of molecules and the morphologies of gels formed through variations in these weak interactions [24].

Different aspects, such as the elastic behavior of solids and the micro-viscous properties of fluids, makes gel a unique class of soft materials. These three-dimensional (3D) networks include the volume of a liquid and entrap it through the surface-tension effect. These fibrous networks may consist of covalent bonds (chemical or polymer gels) or by noncovalent interactions (physical or supramolecular gels) [64].

Sánchez et al., 2010 showed that organic sol–gel chemistry is a way to obtain polyureas. In their process, the key point is the capping of amino-terminated polymers that keep these ends with low reactivity when reacting with isocyanates. The control in the synthesis of polyurea networks results in systems with good thermal stability, low soluble content and more easily tunable mechanical performance with respect to conventional polyurethane, polyurethane/polyurea networks or thermoplastic block copolymers. The position of urea, as cross-linkers, in the hard segments and the preparation of the amino-terminated building block play an important role in the overall elastomeric behavior [30].

Furthermore, CO_2_ gas released during the reaction can be used to induce the formation of a porous structure. The overall chemical reaction process is depicted in Figure 41. Thus, the chemical reactions between isocyanate groups and water could easily form porous polyurea material.

Among the most encouraging route for the creation of PURs is the use of CO_2_ as a carbonyl building block to replace isocyanate. The transformation of CO_2_ directly with diamine to polyurea was first reported by Yamazaki et al., where PUs were obtained from aromatic diamines with CO_2_ by using a stoichiometric amount of phosphate as a catalyst.

More recently, several kinds of PURs were produced by the polymerization of diamine with CO_2_ directly in the absence of any additives, such as water soluble oligourea, macrocyclic oligourea, thermoplastic polyurethane-ureas and PUR hydrophobic gel. The polymerization of diamine with CO_2_ is a form of condensation polymerization, in which water is produced as a by-product, leading to a relatively lower molecular weight as water slows down the reaction kinetics [65].

Martin et al., 2016 proved that the one advantage of the transurethanization approach is its potential to prepare a large range of polyurethanes (without hydroxyl side groups) (Figure 42) since the isocyanate agents can be replaced by dialkyl dicarbamates [66].

However, this technique could not be used directly to prepare cross-linked materials. In this work, they describe an example of a solution to this bottleneck, consisting in the preparation of new renewable non-isocyanate cross-linkable allyl-terminated polyurethanes and polyureas (Figure 43). Allyl groups were chosen because they are easy to introduce, thermally stable and react well under UV to prepare polymer films and coatings [66].

Aqueous polyurethane dispersions (PUDs) with low levels of organic solvent are an alternative to the organic-based dispersions, although these PUDs present inferior resistance to water, surface hydrophilicity and lower mechanical strength. The modification of PUDs with various polydimethylsiloxanes (PDMS), showing increased contact angle and decreased tensile strength with the increase in PDMS content, have been described [68]. When nanoclay is incorporated into polymers, a variety of properties such as flame resistance, mechanical strength, gas barrier and thermal stability are enhanced. The improvement in barrier properties has been reported in polyurethane/clay adhesive nanocomposites [68].

Interfacial polymerization is an effective technique for the synthesis of condensation polymers such as polyurea, polyurethane and polyesters. Using diisocyanate monomers and diamines as the precursors soluble in distinct and immiscible solvents, polymerization takes place at the liquid–liquid interface to form a polymer film. The advantages of interfacial polymerization include rapid reaction rates under ambient conditions, no requirement for reactant stoichiometric balance and a low requirement for reactant purity. The flexible PUR derived from the interfacial polymerization finds many applications in industry, such as encapsulation of pesticides and the micro-encapsulation of drugs and membranes [69].

The introduction of interfacial poly condensation (IP) in PUR research brought a new set of possibilities such as the preparation of bulk polymers, micro/nano-capsules, thin-film composite/nanocomposite membranes (TFCMs), polymer nanocomposites, the surface modification of fibers, micro-unit operations and self-healing materials. IP processes allow for the possibility of the rapid production of polymers (linear or cross-linked) with high and specific molecular weight ranges under normal conditions of temperature and pressure at/or near the interface of two immiscible phases, either liquid–liquid or gas–liquid. The properties of polymer coats are a function of their chemical composition which controls the film thicknesses, crystallinities, molecular weights, degree of cross-linking, mechanical and thermal properties and so on [70].

In the same context, spontaneous precipitation is used for the preparation of PURs and PUs, but for PURs, their high rate of polymerization, synthetic versatility and superior hydrogen bonding capability are advantageous [17,18,19,20,44].

The interfacial condensation (IP) model is shown schematically in Figure 44. The un-ionized part of monomer A diffuses from the aqueous phase through the formed polymer film and reacts with monomer B in a thin reaction zone located on the organic side [71]. For the polyurea system, three phenomena are of importance in such a process in reaction kinetics.

The first is ionic equilibria in the aqueous phase, the second is the diffusion of (unprotonated) m-PDA (meta-phenylene diamine) through the formed polymer film and the third is the chemical reaction of m-PDA with TMC (trimesoyl cloride) on the organic side [72].

Using a π-conjugated urethane-based polymer and π-conjugated urea-based polymer displayed in Figure 45, Ahner et al. compared the emission spectra of their material compared to the π-conjugated system in solution. The emission of the urea polymer 11b is broader and is more strongly emitting at higher wavelengths, whereas the shoulder at 415 nm is more pronounced in urethane polymer 11a. Presumably, the red shifted emission is attributed to dimeric or trimeric aggregates of the chromophores, while the higher energy emission is attributed to weakly interacting monomers [34].

Time- and spectra-resolved emission spectroscopy carried out on the group data allowed one to gain a good understanding of the IP process [34]. For films of both polymers (Figure 45), a biexponential decay was recorded, in contrast to measurements in solution, which fitted with a mono exponential model (Figure 46B). The biexponential decay follows the red shift of emissions with increasing time after the absorption of light (Figure 46C). The bathochromic shift of emission in phenylene ethynylenes are assigned to the co-planarization of the phenyl rings [35]. In the electronic ground state, the phenyl rings are slightly twisted toward each other, but co-planarize in the excited state, resulting in a bathochromic shift in the emission. While the rotation occurs within tens of picoseconds in solution, it is considerably slowed down to hundreds of picoseconds in solid matrices. Thus, the short time constant of the emission decay, τ1, is the time constant of rotation, while the second time constant, τ2, describes the decay of the excited, co-planarized singlet state. For 11a, τ1 is marginally smaller than for 11b. These findings were rationalized by taking the hydrogen bonds into account. In urea polymer 11b, the additional hydrogen bond causes stronger intermolecular interaction, which slightly hinders the rotation of the phenyl rings, and thus impedes the process [34].

The complete set of reactions to form PURs includes the reactions among the oligomers, as well as of the monomers with the oligomers, and results in three types of oligomers of various chain lengths—amine-terminated (A), isocyanate-terminated (B), and amine-isocyanate-terminated (C), with general formulae as given in the table reproduced in Figure 47 [71]. The concentrations of the various species in solution are determined by the output between the rates of reactions that form and consume them, and phase separation [71].

The fact that the oligomers have a strong dependence on the slope of the rate of their consumption on the organic side conditions would indicate (a) that the reaction is on the organic side of the interface and (b) that the reaction is kinetically controlled. Figure 48 shows the effect of the organic phase monomer (HMDI) concentration on the rate of consumption of HMDA. In these runs, the HMDA concentration was the same, but the HMDI concentration was varied to vary R. With an increase in R, i.e., increase in the HMDI concentration in the organic phase, the kinetics is seen to be accelerated [70].

The storage of energy can be associated with Phase Change Materials (PCMs), which can store as well as release energy from or to the surroundings during these changes. The quantity of energy per weight of the material is so large that a lower volume is required by the system to facilitate these energy exchanges. In addition, during the phase change, the temperature remains nearly constant, which is beneficial for the control of the temperature of the surroundings. In the case of interfacial polycondensation (IP) materials such as polyurea, polyurethane, polyester, polyamide and amine resin, these can be used as shell monomers in the interfacial polycondensation process. The core materials are made into droplets. The capsule shell reactive monomers polymerize on the surface of the droplets. When the initially formed oligomers are insoluble at the interface of the droplets, they grow, and a thin monolayer membrane forms around the droplets. The polycondensation causes the monolayer membrane to be a shell and finally leads to the formation of a microscopic shell around the droplets [73].

In particular, urea-based compounds, due to their excellent hydrogen bonding capabilities, are well known for templating the organization of the functional groups that can result in interfacial polycondensation in sheets or lamella structures [34]. PUs synthesized via the traditional method, polycarbonates, polyurethanes and polyamides included are in the table from references [1,18] in Figure 49. Comparing, for example, these polymers with the same chain structures, let us say aliphatic structures, polycarbonates are the most flexible and have no melting temperature because there are no hydrogen bonds between carbonate groups and no crystalline domains can be formed. With the replacement of one oxygen atom in polycarbonates by nitrogen, the polymer changes to polyurethane and mono dentate hydrogen bonds can be formed between urethane groups. This substitution leads to an increase in the polarity and the crystallinity that enhances the rigidity and the melting temperature of the polymer. After the replacement of two oxygen atoms in polycarbonates by nitrogen atoms, the polymer modifies polyureas and bidentate hydrogen bonds can be formed between urea groups, leading to a higher polarity and crystallinity and then a higher rigidity and melting temperature than those of the polyurethanes. The melting temperature and rigidity of the PUs are also higher than those of analogous polyamides. This phenomenon is also ascribed to PUs with two hydrogen-bonded network structures compared with mono dentate hydrogen bonds in polyamides. The hard domains of the PU synthesized by this method are the urea motifs, but the hard domains of the PUs from the traditional methods are the polyisocyanate-based domains (TDI or MDI, for example), leading to no melt state and then limiting their melt processibility [18].

Organic Phase Change Materials (PCMs), when used for concrete or asphalt, should present complementary chemical, physical and thermal properties with these materials, such as an optimum operating temperature, high latent heat fusion/heat capacity, low phase segregation, and safety to humans and the environment, but they still suffer from the drawbacks of supercooling, low heat conductivity, large volume changes, and decomposition upon melting. To overcome these undesirable properties, some studies recommend the esters of long chain carboxylic acids or fatty acids as promising PCM candidates in materials applications [74]. These materials and mixtures of methyl–esters (methyl laurate, methyl palmitate, and methyl stearate) can have a phase transition close to liquid/ice temperatures (2 to −10 °C) or human comfort temperature (~20 °C), added to a relatively high latent heat capacity. They further posit that the charging rate of PCM storage can be improved by using a hermetic encapsulation method via interfacial polymerization, since fatty acid esters are not stable in alkaline environments. A PCM must be encapsulated by a polymeric shell to retain its shape, improve its heat exchange ability and prevent the PCM from leaking and being decomposed during the postprocessing or phase change process [74]. Their compound has two acidic NOH groups that can readily react with n-propylisocyanate to give a 1:2 adduct with high yield and high purity. Thus, the compound can act as a bifunctional monomer and its polymerization reaction with aliphatic and aromatic diisocyanates gave novel polyureas that contain urazole linkages in two dimensions. The polycondensation reaction under microwave irradiation was elected as the best method in this work for the synthesis of polyureas [75]. In addition, their results demonstrate that microwave heating is an efficient method (shorter reaction time and high energy efficiency) for polycondensation reactions [75].

4-Substituted urazoles are five-membered heterocyclic compounds, displaying two NOH acidic protons. The urazole derived from the ene reaction of triazolinediones with alkenes and polydienes has one NOH proton, which is also very acidic. The acidity of this proton has been measured; it has a pKa of 4.71, similar to that of acetic acid. The compounds 1 (Figure 50) have the potential to undergo N-acylation. 4-Substituted-urazole were converted to 1-acyl derivatives by acylation reaction with a series of carboxylic acid anhydrides [76].

Zhang et al., 2009 [77] presented the synthetic process of microencapsulated n-octadecane with polyurea; the formation of the micro-PCMs was implemented by interfacial polycondensation (IP) with the microcapsule shell fabricated on the surface of n-octadecane droplets through polycondensation of the respective monomers. A scheme of this process is shown in Figure 51 [77]. The mixed oil solution consisting of n-octadecane and TDI was dispersed in an aqueous solution by using SMA as an emulsifier, leading to an oil-in-water micro emulsion. The hydrophilic groups of the emulsifier alternatively arrange along its hydrophobic chains, and thus are associated with the water molecules and cover the surface of n-octadecane/TDI mixture oil droplets in an orderly fashion, with hydrophobic chains oriented into the oil droplets and hydrophilic groups out of the oil droplets. In addition, the other requisite monomer, amine, is dropped into the emulsion, and reacts with TDI. The shell-forming reaction is initiated when some of the peripheral isocyanate groups are hydrolyzed at the oil–water interface to form amines, which in turn react with other unhydrolyzed isocyanates. Once the amine is a nucleophilic reagent, it will react with an isocyanate functional group to produce urea. When the initial shell is formed, the water-soluble monomer has to diffuse across the membrane into the oil phase to react with TDI, the oil-soluble monomer, to thicken and strengthen the shell of the microcapsules. As a result, a urea-linked polymeric shell is formed onto the emulsified interface surrounding n-octadecane through the reaction between the amine monomers and TDI [77].

Uniform PU microspheres can be prepared via precipitation polymerization in H_2_O–Acetonitrile mixed solvent with IPDI as the only monomer. According to reference [78], the highest productivity level for the preparation is achieved with IPDI loading of 23 wt% and at 50 °C in the mixed solvent of H_2_O–Acetonitrile at a mass ratio of 20/80. The productivity of highly uniform microspheres obtained was about 22 times higher than that by free radical precipitation polymerization, and practically doubles that previously reported on the same polymerization carried out in H_2_O–acetone [78].

To prepare porous polyurea (PPUR), acetone was used in an optimized process for the preparation of PPUR adsorbent; 90.0 g of H_2_O–acetone mixture at a mass ratio of 3/7 was first put into a 250 mL round bottom flask immersed in a water bath at 30 °C. Under stirring at 300 r/min, 10.0 g of toluene diisocyanate (TDI) was added at a rate of 20 mL/h. The polymerization was allowed to continue for 2 h after TDI addition was completed, followed by drying of the polymer at 70 °C under vacuum to collect the powder product. The yield of the product was 100% through repeated tests due to the step polymerization mechanism involved, as schematized in Figure 52 [79]. To carry out the adsorption of acid fuchsine (AF), representing waste in water, a known amount of PPU was added to a 25 mL glass bottle, to which 20.00 mL of AF aqueous solution (V, mL) was added with known concentration (c_0_, mg/mL) and pre-adjusted pH. The bottle was fixed on a reciprocating oscillator in a 30 °C water bath and shaken at 120 osc/min for 4 h. The contents were centrifuged at 12,000 r/min for 5 min to separate the PPU from the AF solution. This PPUR is also characterized to have excellent desorption and reusability. This work demonstrates that PPUR is an effective absorbent and an attractive candidate for the removal of anionic dyes from wastewater [79].

According to Liang (2009), polyurea microcapsules containing phase change materials were prepared successfully by using interfacial polycondensation. The testing results show that micro-PCMs’ phase change temperature is about 29 °C, the latent heat of fusion is about 80 J, the particle diameter is 20–35 μm, and the particles showed a good property of thermal periodicity. Additionally, with the dry weight analysis, it was possible to obtain a fairly good packing rate for micro-PCMs [73].

Recent efforts have focused on developing non-isocyanate PURs (NIPUT) to replace highly toxic diisocyanates. Mainly, two routes were described: (a) the reaction of bis- or multicyclic carbonates with diamines, and (b) the transurethane polycondensation of diurethanes (Figure 53). The route via transurethane polycondensation is designed to synthesize non-isocyanate thermoplastic polyurethanes (NITPUTs), including amorphous or crystallizable PUT, alternating or segmented poly(amide urethane)s and non-isocyanate thermoplastic polyureas (NI-TPUreas) [80]. These NITPUTs or NI-TPUreas are synthesized from diurethanes prepared previously from the reaction of dimethyl carbonate, diphenyl carbonate, or ethylene carbonate (EC), followed by diamines (DAs), and transurethane polycondensation to synthesize NITPUT or NITPUreas. Dai et al. recently reported a method to synthesize NI-TPUreas through the direct solution-free polycondensation of DAs with diphenyl carbonate [80].

Supramolecular gels are good candidates for soft, stimuli-responsive materials, as they combine the elastic behavior of solids with the micro viscous properties of fluids. The dynamic networks of fibers in supramolecular gels are suggestive of the cytoskeleton of a cell and provide scaffolds to implement function. When gels are made responsive to stimuli, these mechanical properties can be controlled. Sol–gel transitions also create opportunities to immobilize molecules inside the gel’s cavities and to release them on demand. To establish selective responsiveness, suitable recognition sites are required, influencing the properties of the fiber network depending on the presence of the stimulus [64].

## 6. Mechanical Aspects

Copolymer polyols, that is, polyols with polymeric fillers, include: (a) conventional copolymer polyols where the filler particles are copolymers of styrene and acrylonitrile; (b) PHD polyols, where the filler particles are polyureas; (c) PIPA polyols, where the filler particles are polyurethanes; and (d) epoxy dispersion polyols, where the filler particles are cured epoxy resins. In these copolymers, the filler particles usually have an aspect ratio (A_f_) close to one, where the A_f_ is defined as the ratio of the lengths of the longest and shortest dimensions for ellipsoid particles. Accordingly, A_f_ ~ 1 characterizes a particle of nearly spherical shape, including both spheres (A_f_ = 1) and irregular shapes with no major shape anisotropy. It is known that in a filler where A_f_ ~ 1 provides the least effective type of reinforcement, the magnitude of the reinforcement is quantified by the ratio of the tensile (Young’s) moduli of the filled and unfilled polymers. The reinforcing efficiency of filler particles possessing a given set of mechanical properties, when incorporated in a matrix material at a given volume fraction, rapidly increases with the increasing shape anisotropy of the particles, and eventually, asymptotically approaches the continuous fiber limit for prolate particles and the infinitesimally thin disk limit for oblate particles (platelets). The reactions of urea with 1,6-hexanediamine at 150 °C in a polyol continuous phase in the presence of a stabilizer for the resultant particles produce low-molecular-weight oligomers of urea-terminated poly(1,6-hexamethyleneurea) with an M_n_ in the range 500–700 g moL^−1^. This is assumed to involve a polymerization/precipitation mechanism in which the molecular units are held together by hydrogen bonding in a macrostructure which separates as a stable dispersed phase. This macrostructure has a spiral fiber bundle morphology, a particle size distribution of ~1 to 10 µm and aspect ratios from 6 to 20. The urea-terminated poly(l,6-hexamethyleneurea) particles are highly crystalline thermoplastics with a melting point of ~270 °C and are only soluble in strong acids, where the macroparticles dissociate into their molecular units. High-molecular-weight poly(1,6_hexamethyleneurea) has been synthesized in bulk by other methods and is reported to have a melting point in the range of 270–300 °C. However, urea-terminated poly(1,6-hexamethyleneurea) oligomers with spiral fiber bundle morphologies have not been reported [81].

Amini et al., 2010 discussed the concept that the stiffness of polyurea increases significantly when subjected to increasing pressure, and when confined polyurea is loaded in compression, its stiffness can be enhanced by 10–20-fold [82]. This leads to polyureas with better impedance matching steel plates, thus causing more energy to be transmitted to the plate, and subsequently initiating the damage factors on the plate [8,82]. The mechanical properties of PUs highly depend on temperature, pressure and the rate of deformation. The glass transition temperature, Tg, of PU is around −50°, which is conveniently low compared to its standard uses temperature. Roland et al., 2007 reported stress–strain measurements for an elastomer PU in uniaxial tension over a range of strain rates from 0.06 to 573 s^−1^ [83].

In parallel, Sarva et al., 2007 reported the uniaxial compression stress–strain behavior of a representative polyurea and a representative polyurethane over a wide range of strain rates, from 0.001 s^−1^ to 10,000 s^−1^ [84]. They compared their data to other researchers’ data and observed that PU undergoes a transition from a rubbery-regime behavior at low rates to a leathery-regime behavior at highest rates. Above the glass transition temperature, polyurea has a nearly elastic volumetric response and a viscoelastic shear response at moderate pressures and strain rates. At room temperature, PU is highly elastic, resistant to abrasion and can undergo up to 800% elongation prior to rupture [7].

PUs exhibit unique viscoelastic properties that depend on pressure, temperature and strain rate. Further, the micro-phase segregation of hard and soft domains in combination with extensive hydrogen bonding allows PU-copolymers’ mechanical stiffness and toughness to be chemically tailored [9]. These copolymers exhibit unique properties as a result of their phase-separated morphology, which is due to the segregation of dissimilar blocks being restricted under a 100 nm hard domain size. The soft segments form a continuous matrix reinforced by the hard segments that are randomly dispersed as nanodomains. Typically, the long and soft diamine blocks form a flexible matrix. The multifunctional hard segment domains serve as both physical cross-links and as reinforcing fillers. High mechanical toughness is a result of the extensive intermolecular hydrogen bonding in polyurea hard domains [85].

Importantly, polymer nanoencapsulation may reinforce the skeletal framework to the point where it can withstand the capillary forces exerted around the receding meniscus of evaporating low vapor pressure solvents (e.g., pentane), allowing it to be dried under ambient pressure [86].

The formation process of porous PUs (HPUs) is illustrated in Figure 54. First, the hydrolysis of naphthalene diisocyanate (NDI) leads to carbamic acid (Figure 54, step 1at) that decomposes into naphthalene diamine (NDA) and CO_2_ in a fast step (step 1b). NDA is more reactive than H_2_O and reacts with NDI quickly, yielding polyurea (step 2). In the process, the generated CO_2_ gas acts as a foaming agent and creates macropores in the polymers. These processes have the same mechanism as in the production of polyurethane foam in the industry. After the H_2_O is consumed completely, the remaining isocyanate groups can react with urea groups on the molecular chains of polyurea at a high temperature (150 °C), generating the hyper-cross-linking structure (step 3) with meso- and micropores. Eventually, hierarchically structured porous PUs are formed [87].

The stress–strain behavior of both polyurea and polyurethane are rate dependent [9]. While researching the high-strain rate mechanical behavior of polyurea, Roland et al. (2007) demonstrated the influence of stoichiometry on the low strain rate response. It was shown that a 5 to 10% variation in chemistry could lead to dramatic changes in the mechanical properties. Increased isocyanate content resulted in increased yield stress and decreased failure strain. The results demonstrated that increasing the amount of isocyanate component is necessary to drive the cross-linking reaction towards completion [85].

There continues to be intense research directed toward the preparation and characterization of new organic polymeric materials for second-order nonlinear optics. Thin films of these materials potentially could be used in optoelectronic devices because they have very attractive mechanical and electronic properties, such as femtosecond response times, high polarizability and processibility. These polymeric materials are also far less expensive than conventional nonlinear optical (NLO) inorganic materials. By the application of electric field poling such as corona poling, the chromophores within these polymers possessing high molecular polarizabilities can be effectively aligned to produce NLO thin films. These films have been shown to be effective in electro-optic (E–O) modulation and second harmonic generation (SHG). One of the greatest issues to address for poled polymeric NLO materials is the stability of the orientation of the NLO molecules. In this paper, we describe the synthesis and NLO properties of a side-chain polyurethane with excellent long-term stability [88].

## 7. Conclusions

Polyureas constitutes a special class of polymers, as their analogs are polyurethanes. Whereas polyurethanes have a considerable number of applications as well as a large quantity of theoretical and practical literature, polyureas are still relatively less researched. The variety of synthetic approaches confers to, for practical purposes, a selectable choice that depends on available resources. It can be made from harmful reactants such as isocyanides to atmospheric CO_2_ and N_2_ capture, or other synthetic routes. One good feature of PURs is their more controllable synthetic processes given the difference in reactivity of diamines towards that of dialcohols. The state-of-the-art of materials processing include polyureas, which have applications in various fields. The presence of urea moieties in PUs allows the formation of several types of polymer bundles and composites based on H-bond-directed growth. This, as well several types of gelators for sol–gel processing, is another feature of these compounds. In short, a large perspective is an attribute of polyureas. One strong feature of the urea group is the adduct formation with strong acids such as phosphoric and sulfuric, which creates possibilities for the use of polyureas in transport and storage.

## Figures and Tables

**Figure 1 polymers-13-04393-f001:**
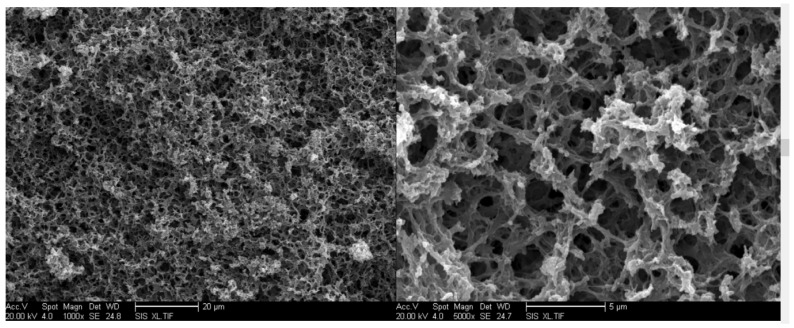
SEM images of porous polyurea monolith at centrifugal speed of 3000 rpm in low and high magnification at 1000× and 5000× magnification. From reference [16].

**Figure 2 polymers-13-04393-f002:**
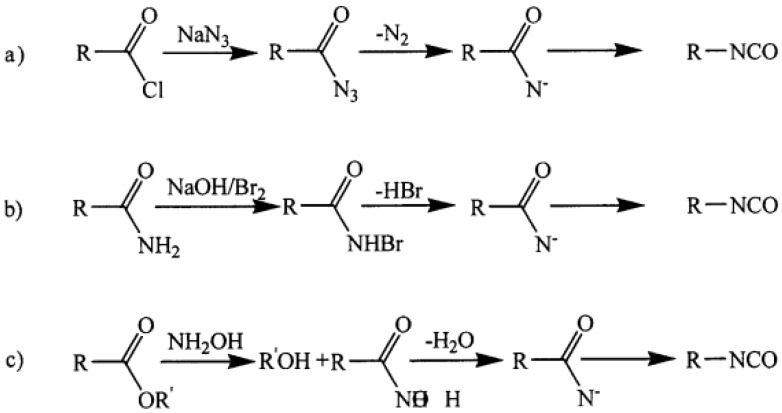
Laboratory-scale synthesis of isocyanates: (**a**) Curtius, (**b**) Hoffman, and (**c**) Lossen rearrangements. From reference [24].

**Figure 3 polymers-13-04393-f003:**
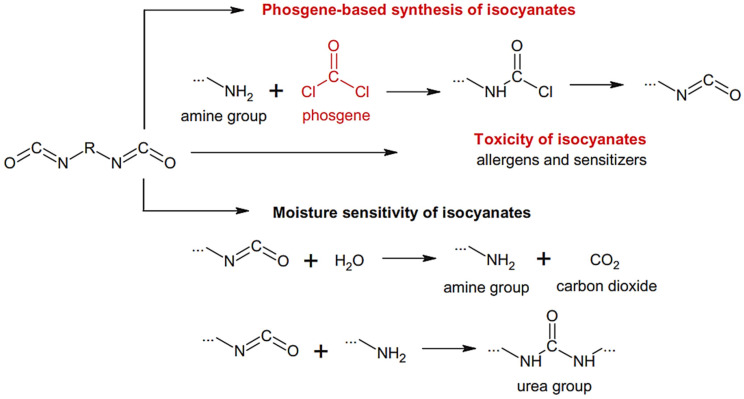
Synthesis of phosgene-based isocyanates. From reference [27].

**Figure 4 polymers-13-04393-f004:**
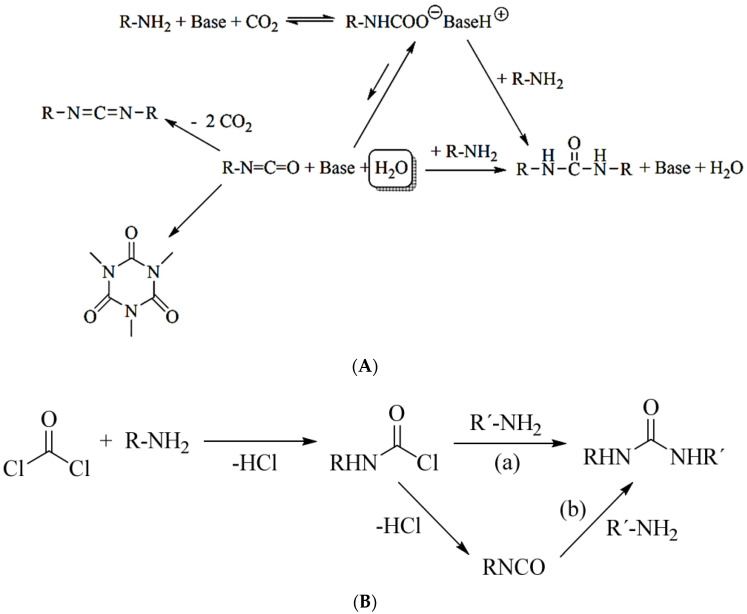
(**A**) Reaction pathways of a primary amine and CO_2_ in presence of excess base. (**B**) Synthesis of urea derivatives: direct (**a**) and indirect (**b**) method. From reference [31].

**Figure 5 polymers-13-04393-f005:**
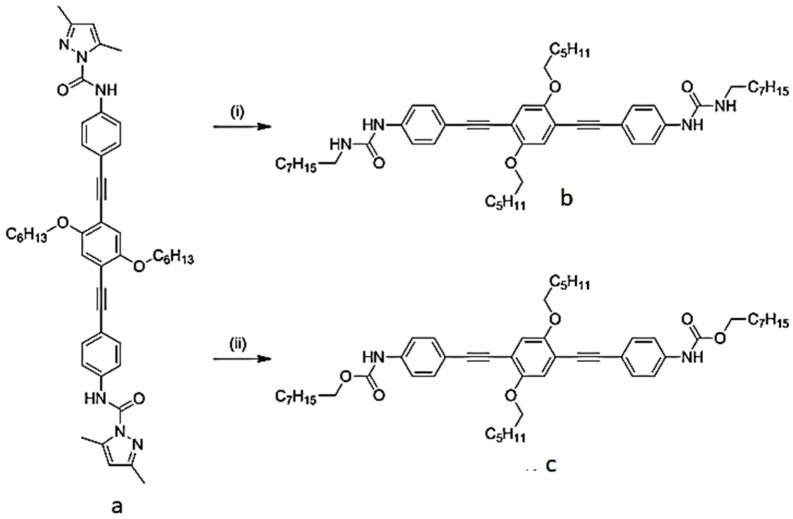
Schematic representation of the synthetic procedure of the model compounds (**a**–**c**). Reagents and conditions: (i) 1-Octylamine/*N*,*N*′-dimethylformamide/130 °C; (ii) 1-octanol/*N*,*N*′-dimethylformamide/130 °C. From reference [34].

**Figure 6 polymers-13-04393-f006:**
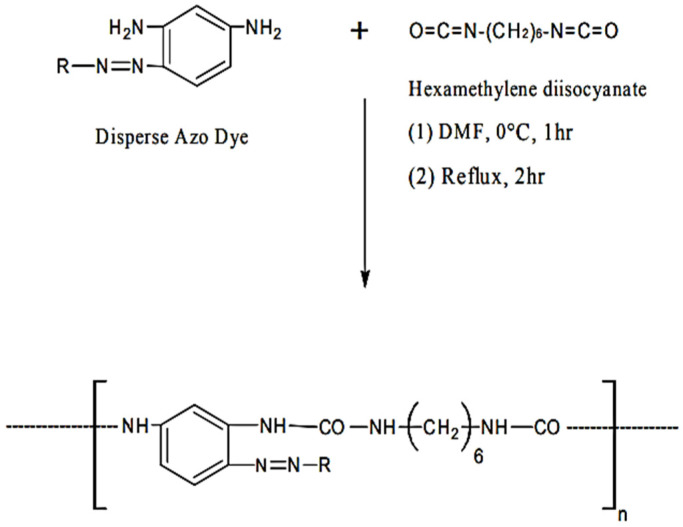
Synthesis of polyureas based on azo disperse dyes. From reference [19].

**Figure 7 polymers-13-04393-f007:**
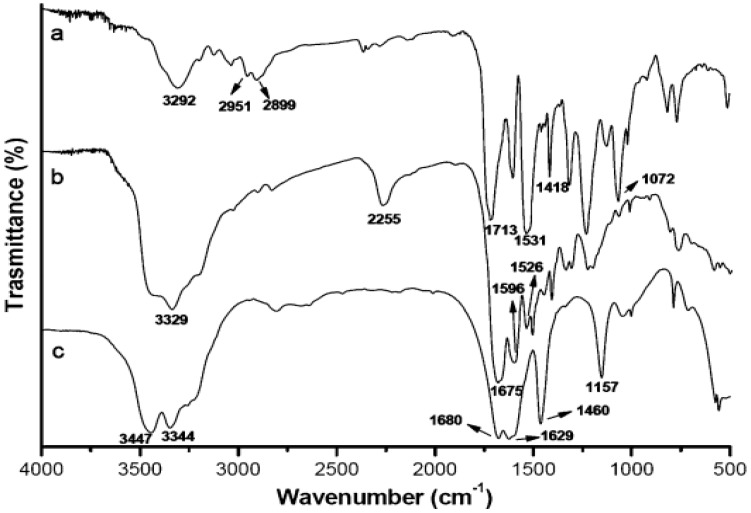
FTIR spectra of PNG (**a**), PN (**b**), and urea (**c**). From reference [37].

**Figure 8 polymers-13-04393-f008:**
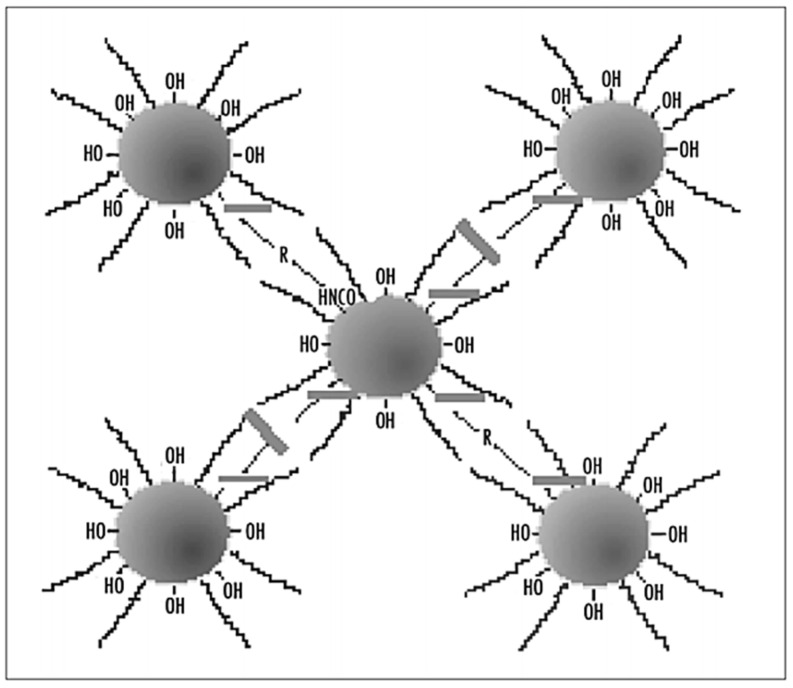
Schematic illustration of hyper-branched polyurethane dispersion. From reference [39].

**Figure 9 polymers-13-04393-f009:**
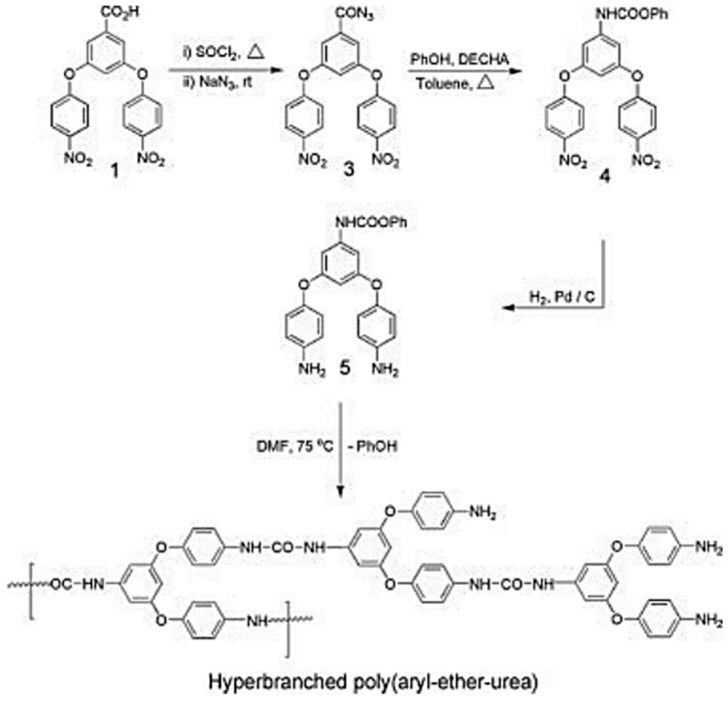
Synthesis of AB_2_-type blocked isocyanate monomer and corresponding hyperbranched poly(aryl-ether-urea). From reference [41].

**Figure 10 polymers-13-04393-f010:**
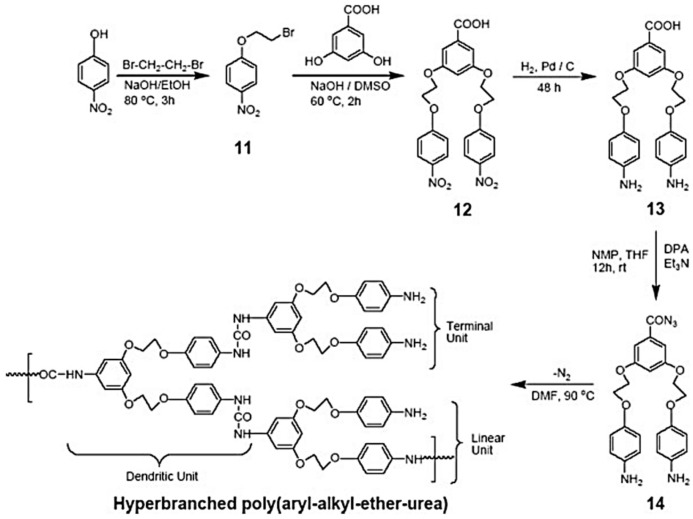
Synthesis of AB_2_-type azide monomer corresponding hyper-branched poly(aryl-alkyl-ether-urea). From reference [41].

**Figure 11 polymers-13-04393-f011:**
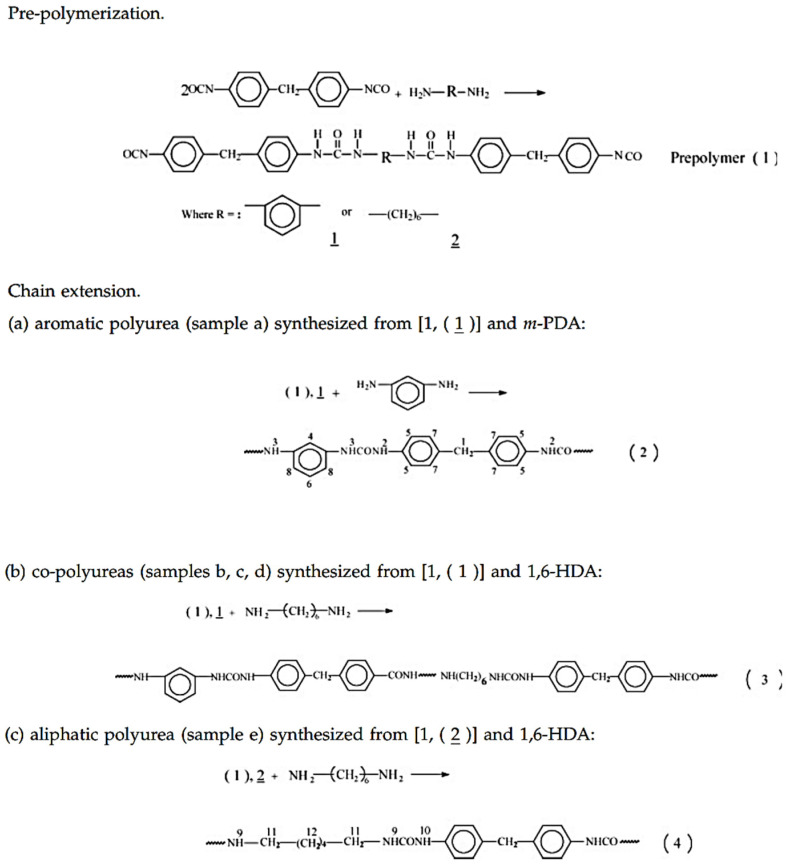
Typical polyurea polymerization reactions, adapted from reference [42].

**Figure 12 polymers-13-04393-f012:**
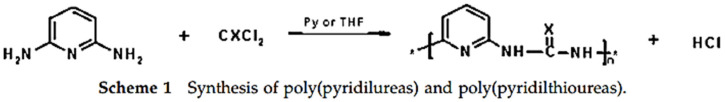
Synthesis of poly(pyridilureas) and poly(piridilthioureas). From reference [47].

**Figure 13 polymers-13-04393-f013:**
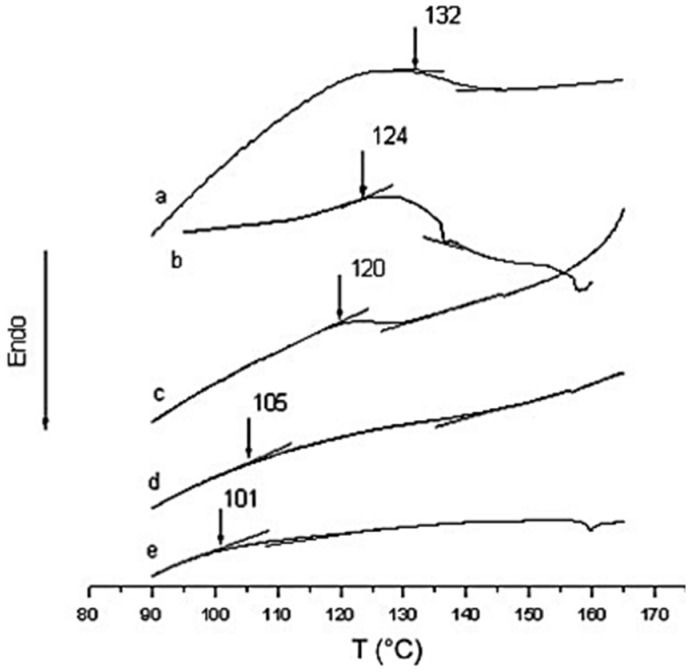
DSC curves of the polyurea samples ((**a**) aromatic polyurea; (**b**–**d**) co-polyureas; (**e**) ali-phatic polyurea). From reference [42].

**Figure 14 polymers-13-04393-f014:**
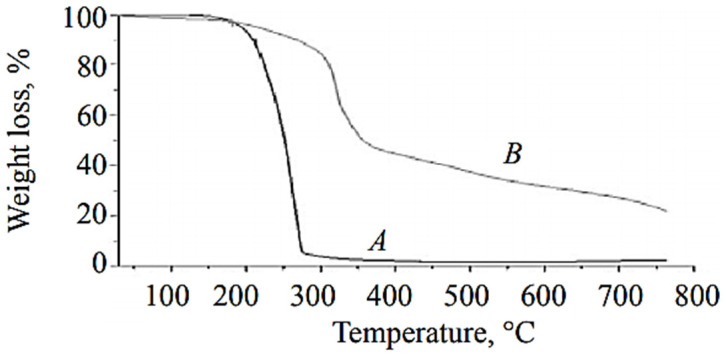
Comparative TGA curves for (**A**) pure metribuzin as a core and (**B**) polyurea microcapsules. From reference [49].

**Figure 15 polymers-13-04393-f015:**
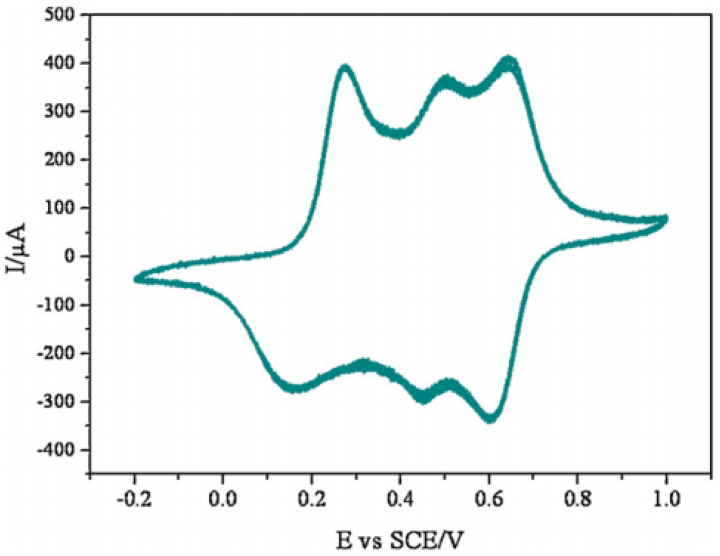
CV measurements of EPU measured in aqueous H_2_SO_4_ (1.0 M) at a scan rate of 50 mVs^−1^. From reference [51].

**Figure 16 polymers-13-04393-f016:**
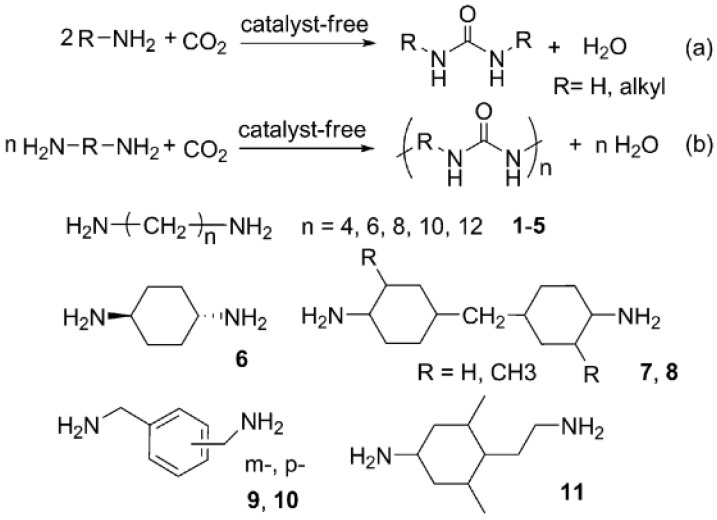
Synthesis of urea, urea derivatives or polyureas from different kinds of amines and CO_2_. (**a**,**b**) are the products from monoamine and diamine reaction. From reference [18].

**Figure 17 polymers-13-04393-f017:**
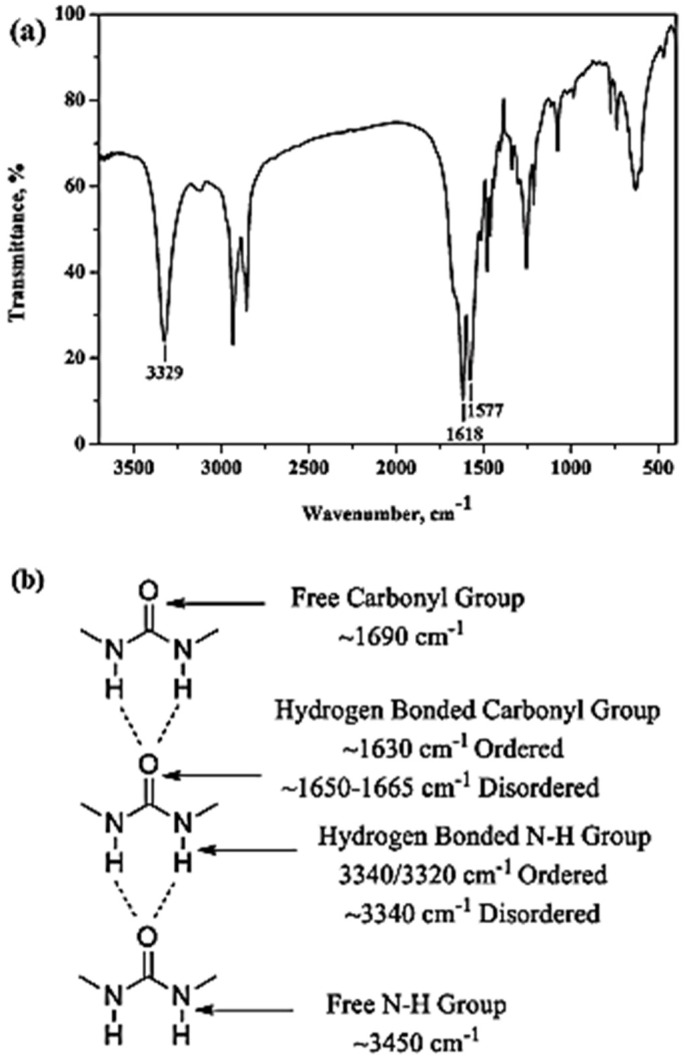
(**a**) The FTIR spectrum of the solid product of the reaction of HDA with CO_2_. (**b**) Band assignments for the polyurea carbonyl and N-H stretching modes. From reference [48].

**Figure 18 polymers-13-04393-f018:**
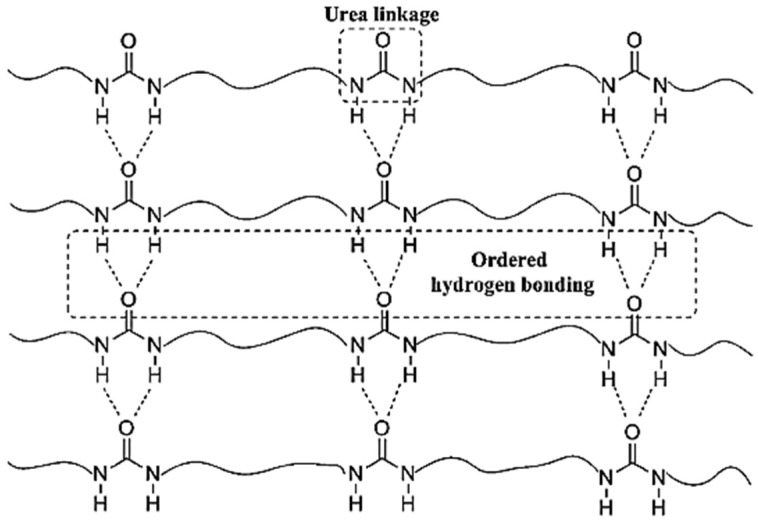
The network structure of the polyurea-HDA. From reference [48].

**Figure 19 polymers-13-04393-f019:**
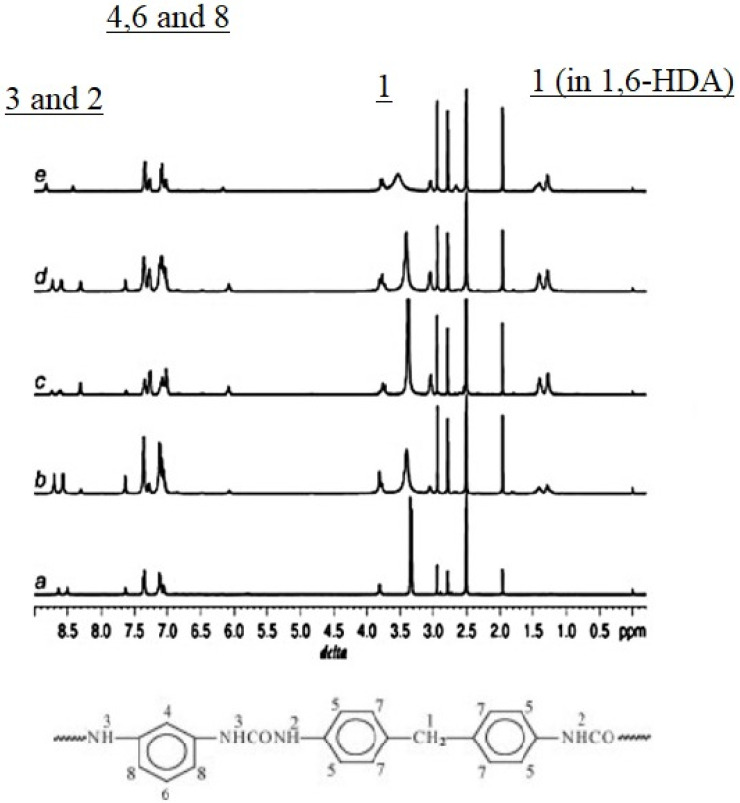
H-NMR spectra of the polyurea samples. From reference [42].

**Figure 20 polymers-13-04393-f020:**
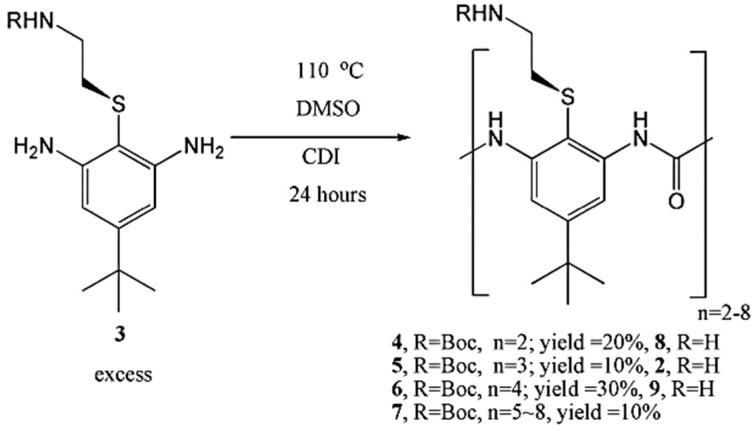
Synthesis of urea oligomers. From reference [54].

**Figure 21 polymers-13-04393-f021:**
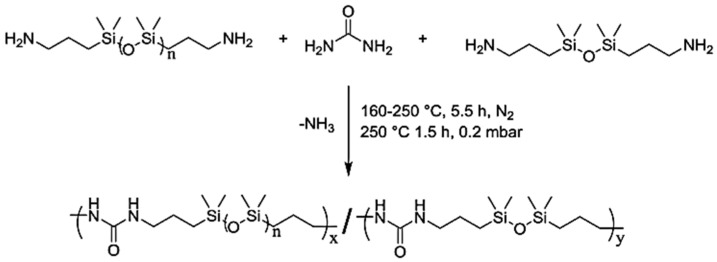
Synthesis of isocyanate-free, segmented, poly(dimethyl siloxane) (PDMS)-containing polyureas with bis(3-aminopropyl)tetramethyldisiloxane (BATS) as a chain extender via melt polycondensation in the absence of solvent and catalyst (poly(PDMS1.7kU)_x_-co-poly(BATSU)_Y_). From reference [23].

**Figure 22 polymers-13-04393-f022:**
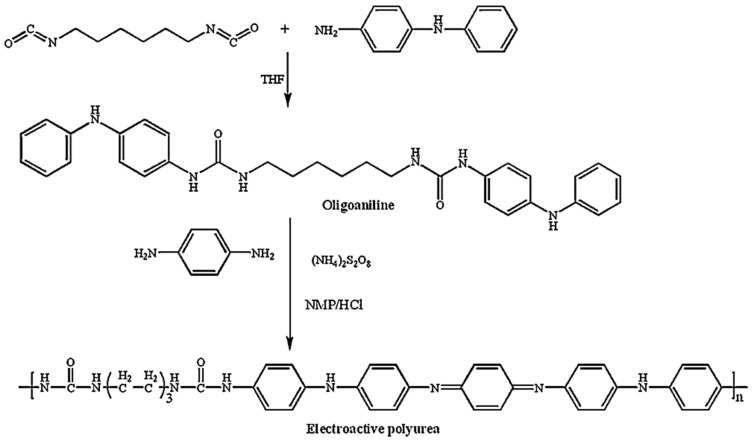
Synthesis route of the oligoaniline and EPU. From reference [51].

**Figure 23 polymers-13-04393-f023:**
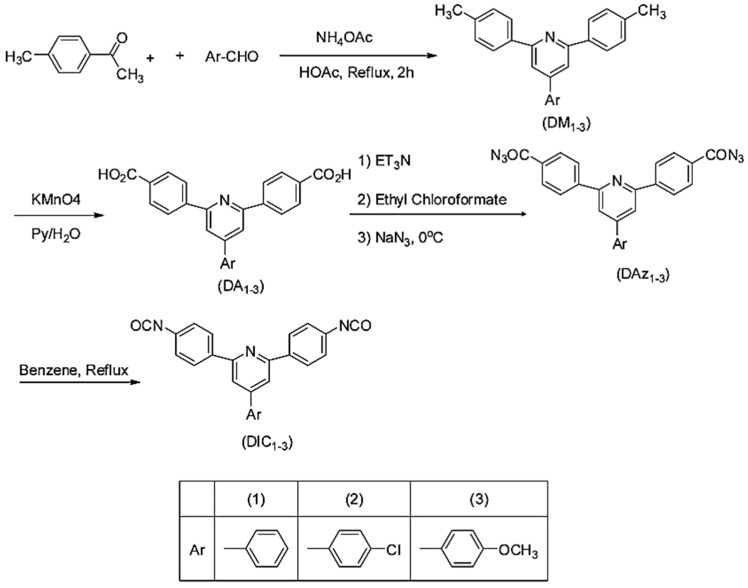
Synthesis of diacids and related diisocyanate monomers. From reference [55].

**Figure 24 polymers-13-04393-f024:**
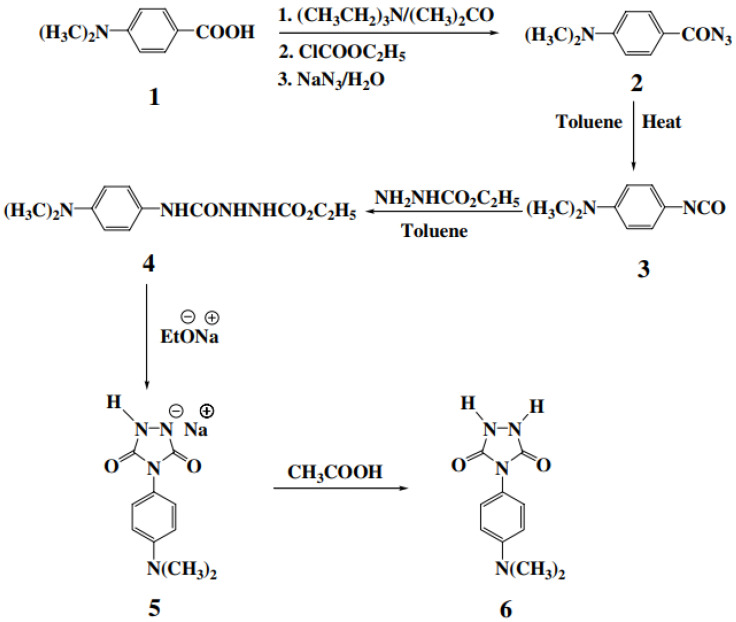
Synthesis of monomer. From reference [56].

**Figure 25 polymers-13-04393-f025:**
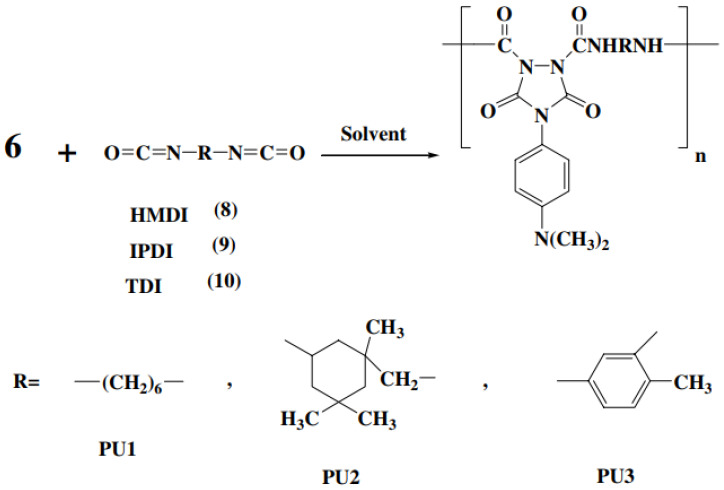
Polycondensation reactions of monomer 6 with diisocyanates. From reference [56].

**Figure 26 polymers-13-04393-f026:**
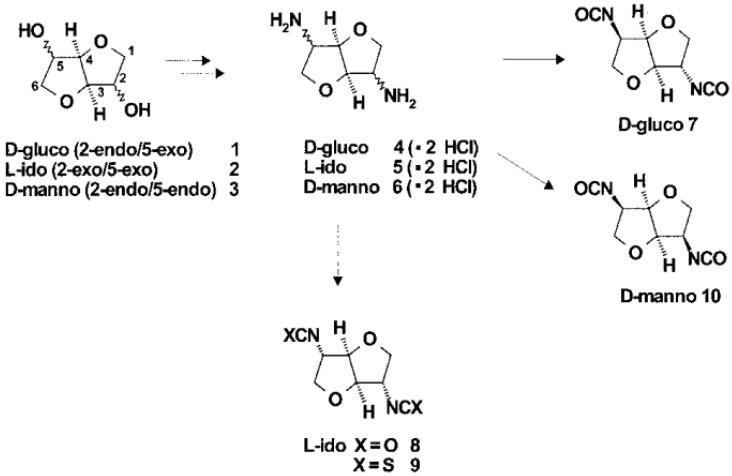
Monomer synthesis. From reference [57].

**Figure 27 polymers-13-04393-f027:**
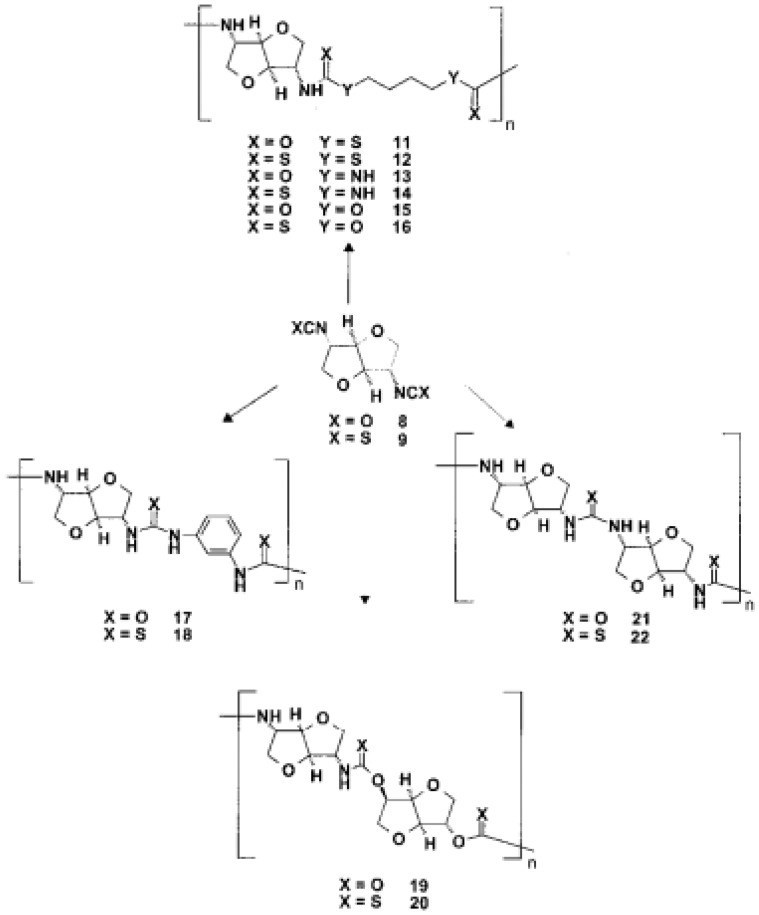
Synthesis of poly(thio)urethanes and poly(thio)ureas from the L-ido monomers 8 and 9. From reference [57].

**Figure 28 polymers-13-04393-f028:**
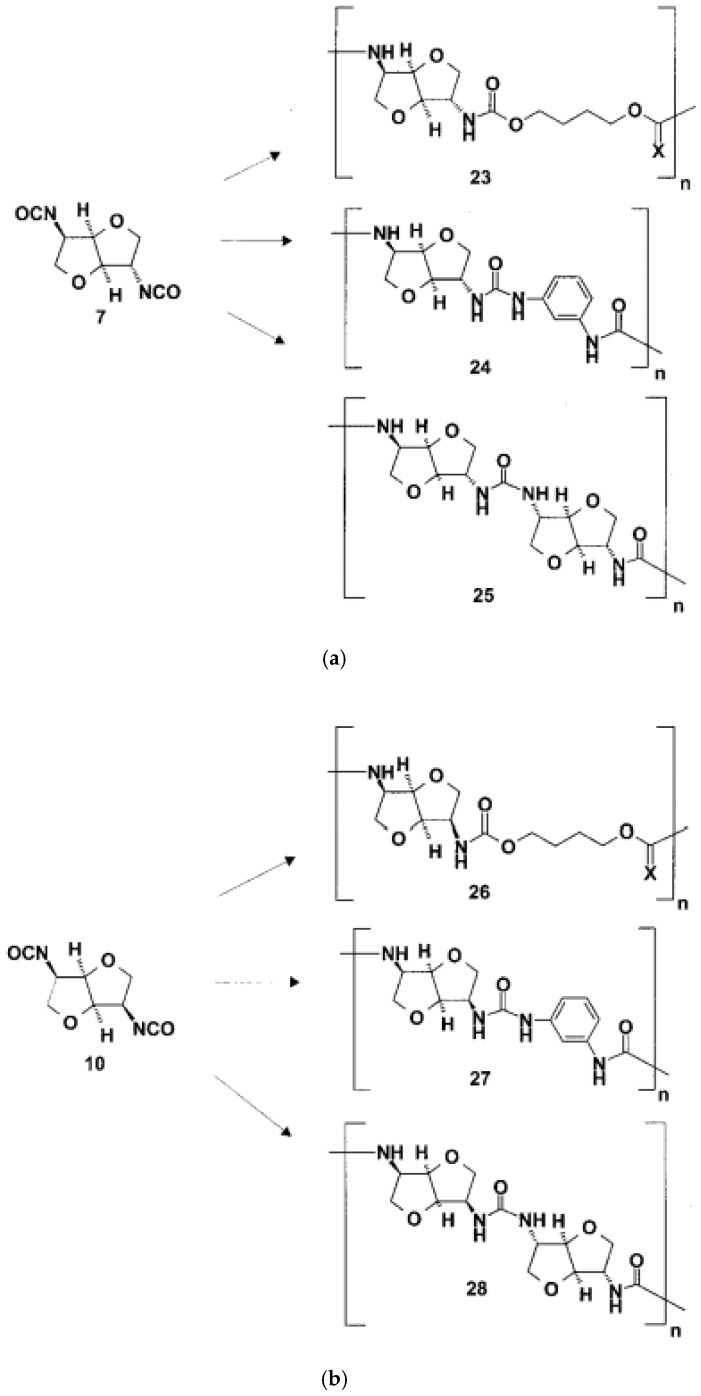
(**a**) Synthesis of polyurethane and of polyureas from the D-gluco monomers 7. From reference [57]. (**b**) Synthesis of polyurethane and of polyureas from the D-manno monomers 10. From reference [57].

**Figure 29 polymers-13-04393-f029:**
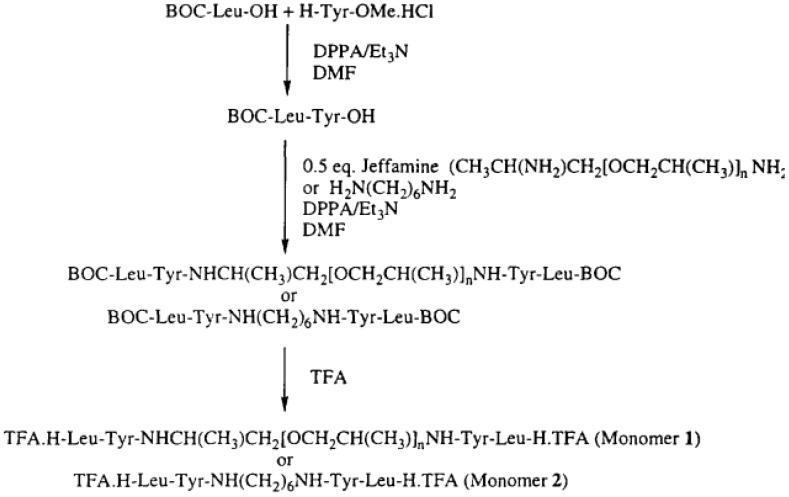
Synthesis of monomers. From reference [58].

**Figure 30 polymers-13-04393-f030:**
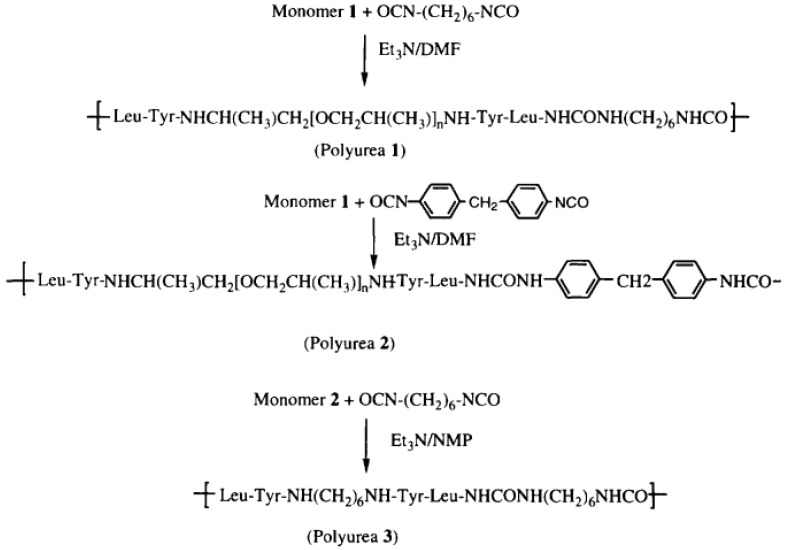
Synthesis of polyureas. From reference [58].

**Figure 31 polymers-13-04393-f031:**
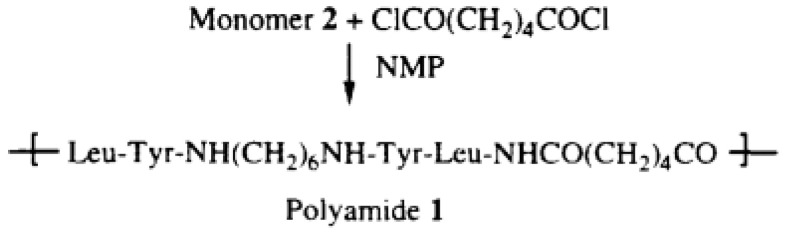
Synthesis of polyamides. From reference [58].

**Figure 32 polymers-13-04393-f032:**
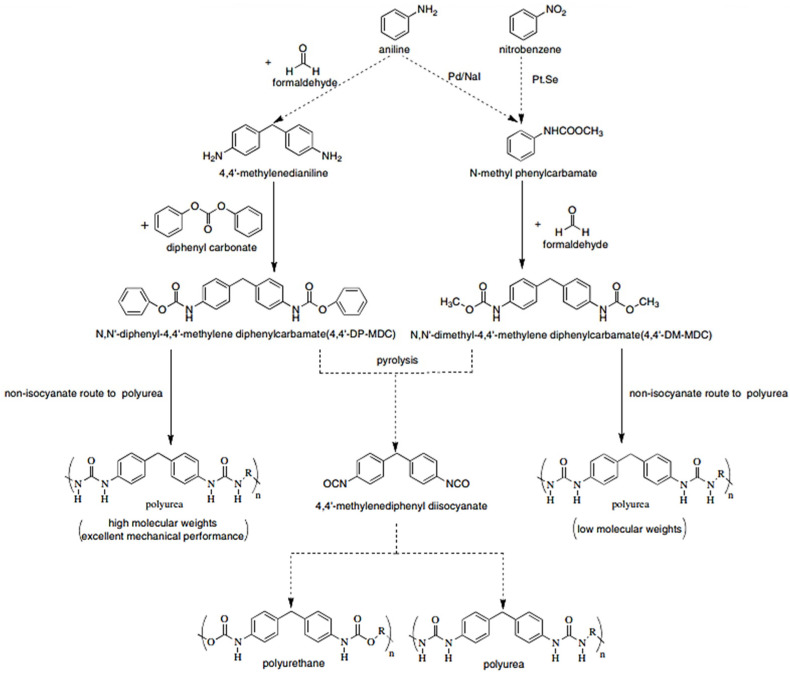
Overall scheme of NIR route of polyurea (the solid-line portions are carried out successfully in the study). From reference [17].

**Figure 33 polymers-13-04393-f033:**
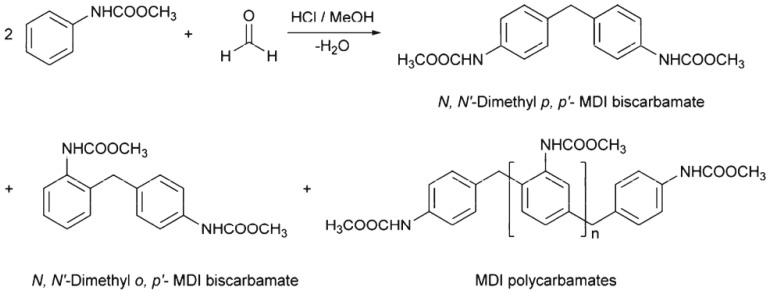
Condensation of N-methyl phenylcarbamate with formaldehyde (synthesis of 4,4′-DN-MDC). From reference [17].

**Figure 34 polymers-13-04393-f034:**
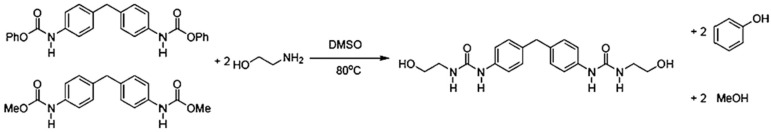
Synthesis of 4,4′diphenylmethanbis-[(2-hydroxyethyl)urea].(Model trans-ureation reaction). From reference [17].

**Figure 35 polymers-13-04393-f035:**
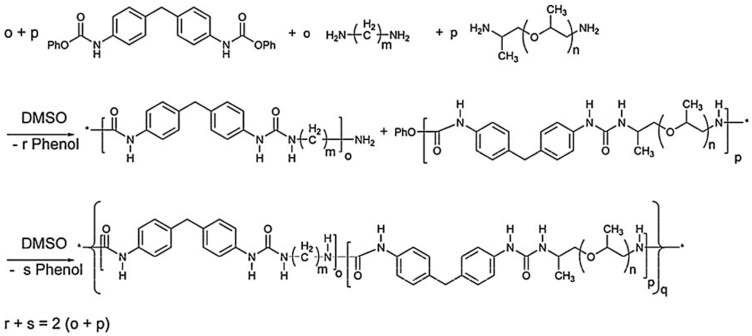
Polyurea elastomer synthesis from 4,4′-DP-MDC in DMSO/80 °C. From reference [17].

**Figure 36 polymers-13-04393-f036:**
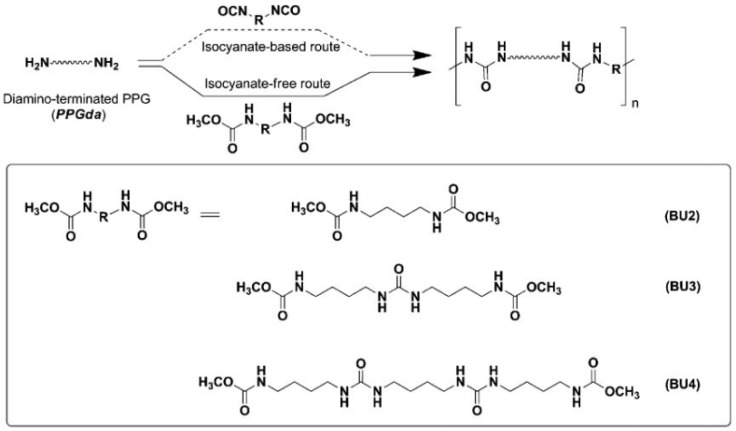
Synthesis of PUs from dicarbamates and diamino-terminated PPG via an isocyanate-free route. From reference [60].

**Figure 37 polymers-13-04393-f037:**
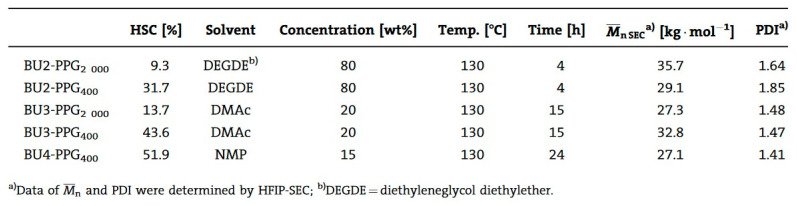
Table copied from reference [60]. Results of polymerizations of dicarbamates with PPGda. From reference [60].

**Figure 38 polymers-13-04393-f038:**
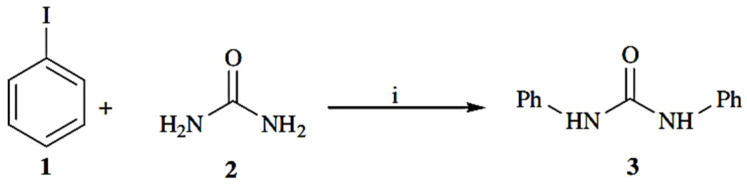
CuI, 1,2-diaminocyclohexane, K_3_PO_4_, DMF, 80 °C, 24 h. From reference [61].

**Figure 39 polymers-13-04393-f039:**
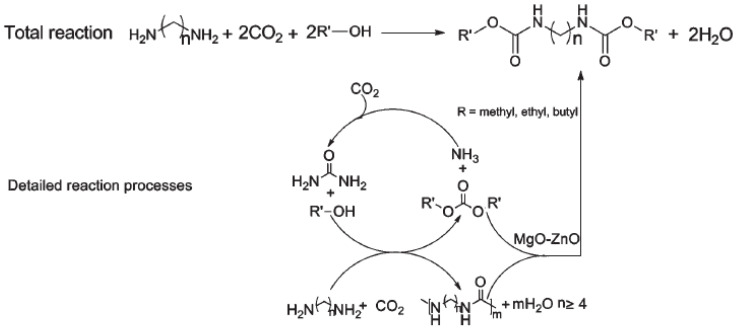
Synthesis of N-substituted dicarbamates from diakyl carbonates and polyurea derivatives based on diamines and CO_2_. From reference [48].

**Figure 40 polymers-13-04393-f040:**
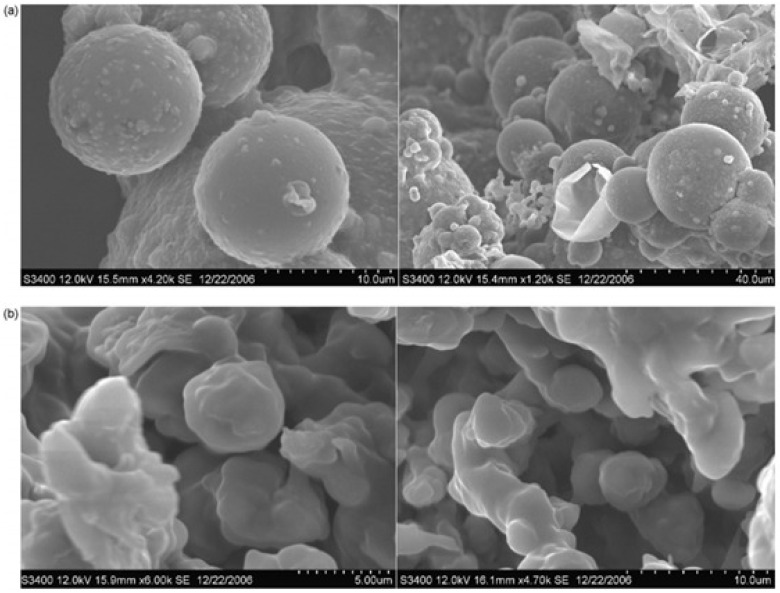
SEM pictures of the microcapsules formed using (**a**) SLS as surfactant and (**b**) Tween-85 as surfactant. From reference [62].

**Figure 41 polymers-13-04393-f041:**
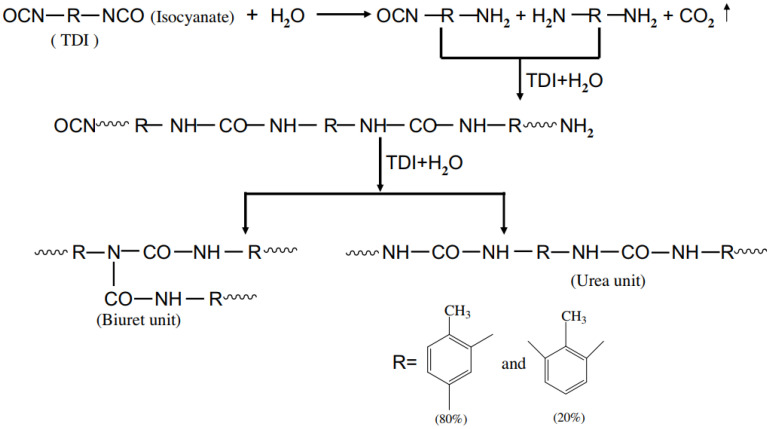
The chemical reactions process between the isocyanate groups and water. From reference [16].

**Figure 42 polymers-13-04393-f042:**
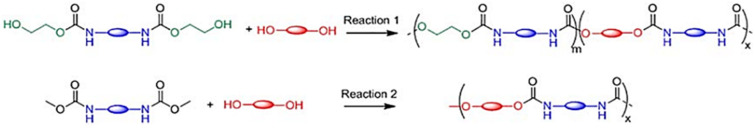
Two reactions related to transurethane polycondensation. From reference [67].

**Figure 43 polymers-13-04393-f043:**
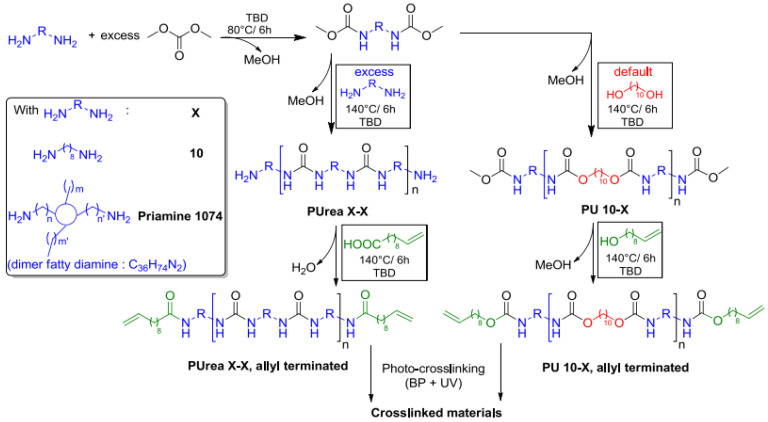
Synthesis of renewable allyl terminated polyurethanes and polyureas. From reference [66].

**Figure 44 polymers-13-04393-f044:**
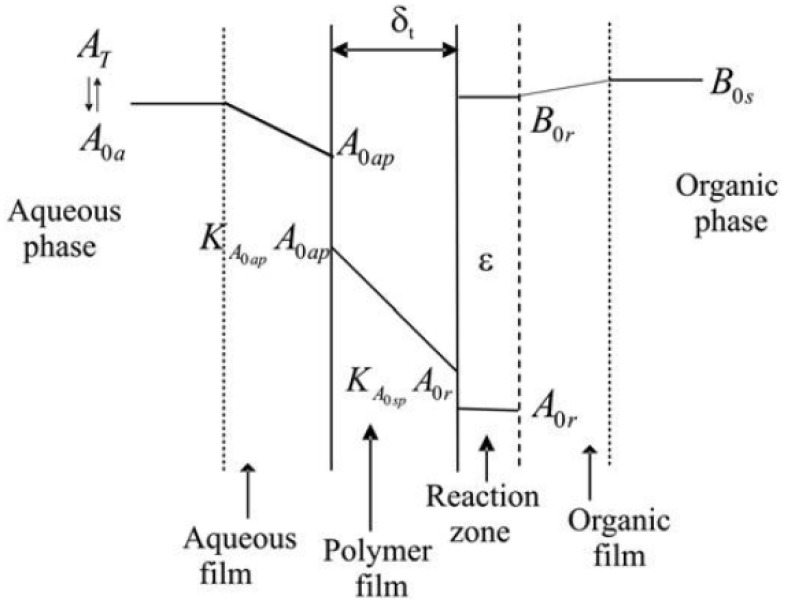
Schematic showing the concentration profiles across the polymer film during IP reaction. From reference [71].

**Figure 45 polymers-13-04393-f045:**
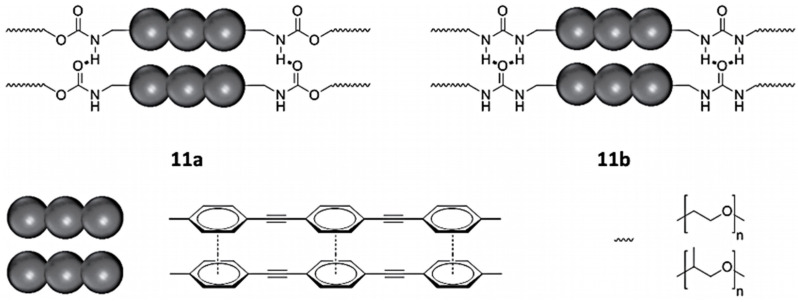
Schematic representation of supramolecular interactions of π-conjugated urethane-based polymer (**11a**) and π-conjugated urea based polymer (**11b**). From reference [34].

**Figure 46 polymers-13-04393-f046:**
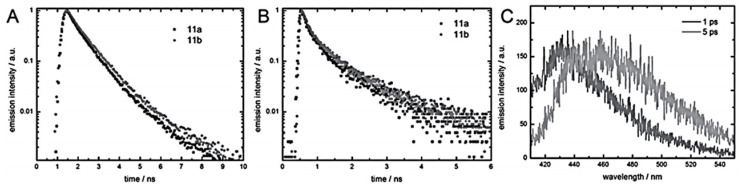
Time-resolved emission data of π-conjugated urethane-based polymer 11b (gray) recorded after 380 nm excitation in (**A**) chloroform and of (**B**) thin films. All decay curves have been spectrally integrated between 420 and 500 nm and normalized to their respective intensity maximum. (**C**) Integrated time-resolved emission spectra of 11a within 1 ns after excitation (gray) and between 1 and 5 ns after excitation (light gray). From reference [34].

**Figure 47 polymers-13-04393-f047:**
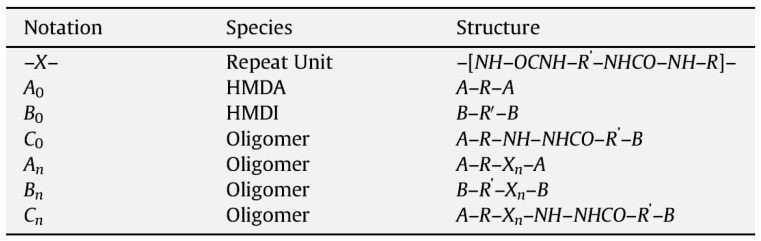
Table copied from reference [71]. General formulae for the different oligomeric and monomeric species that form during the reaction. The polyurea system is taken as the example here (A-: NH_2_-: B-: NCO-; R:(CH_2_)_6_; R′(CH_2_)_6_). From reference [71].

**Figure 48 polymers-13-04393-f048:**
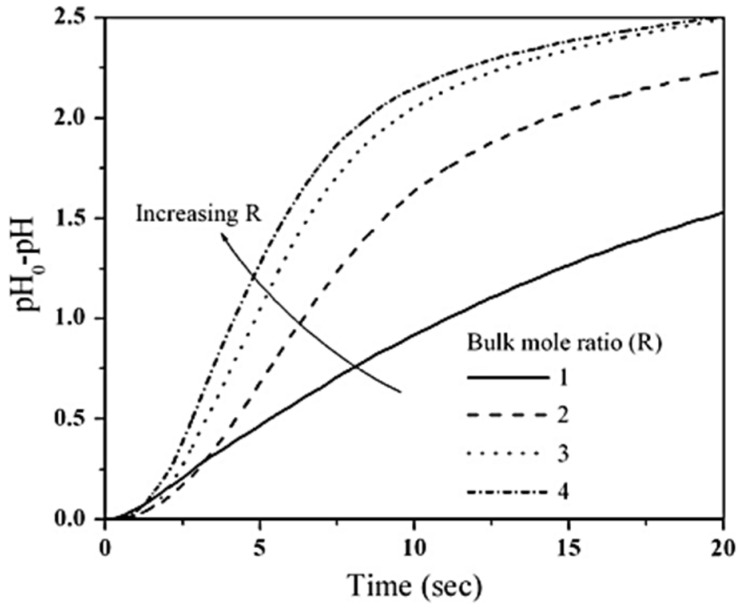
Effect of the monomer mole ratio of HMDI to HMDA (R) on process kinetics (n_L_/V_d_ = 0.545 kmol/m^3^). From reference [70].

**Figure 49 polymers-13-04393-f049:**
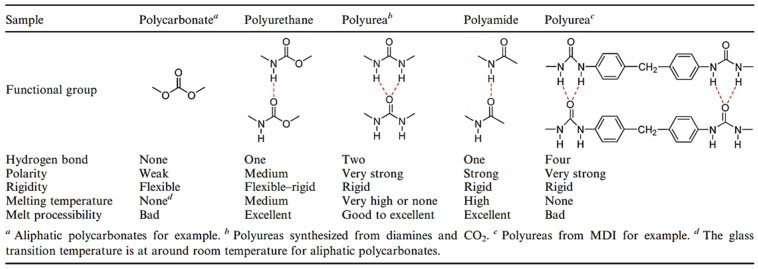
Comparison of some polymers with similar functional groups. From reference [1].

**Figure 50 polymers-13-04393-f050:**
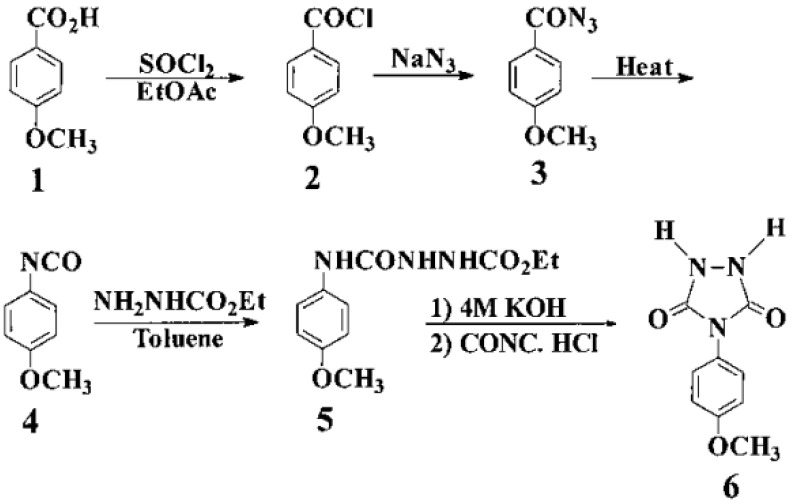
Synthesis of ureazole monomers. From reference [76].

**Figure 51 polymers-13-04393-f051:**
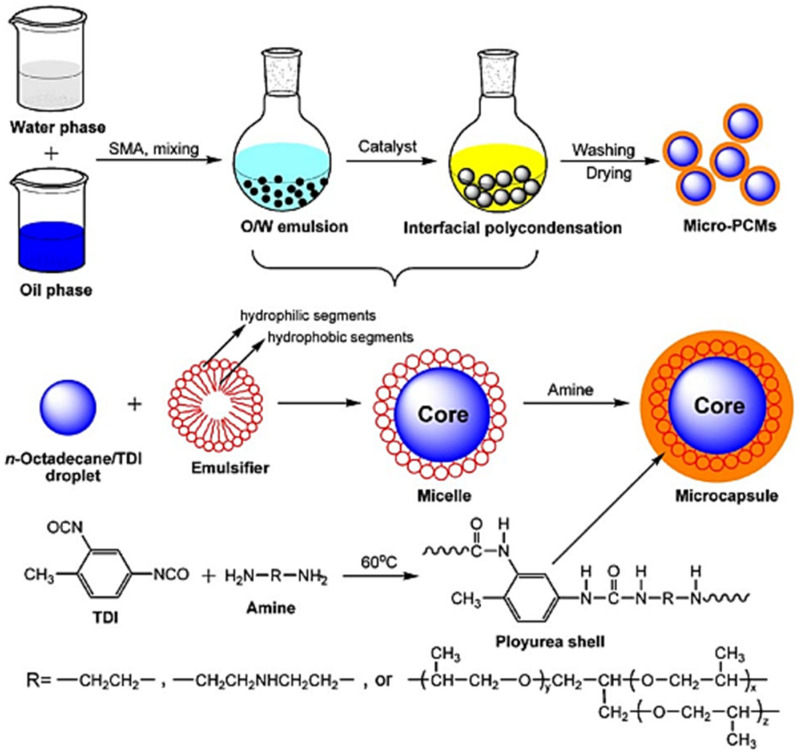
Schematic formation of the microencapsulated n-octadecane with the polyurea shells containing different soft segments through interfacial polycondensation. From reference [77].

**Figure 52 polymers-13-04393-f052:**
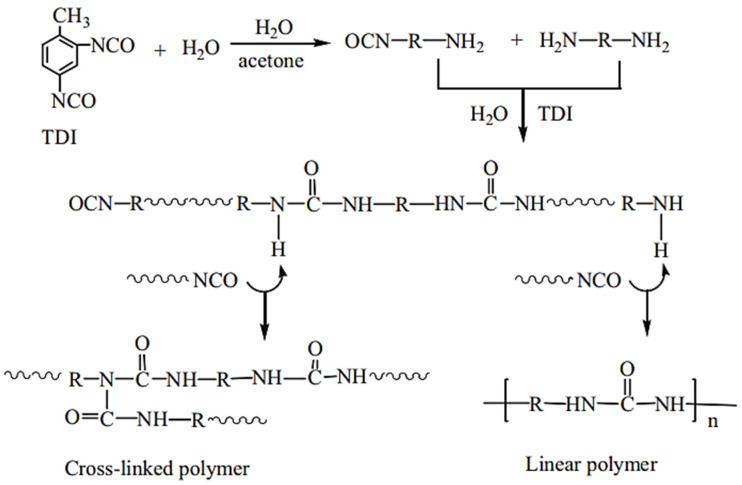
Schematic presentation of porous polyurea absorbent through precipitation polymerization of TDI and water. From reference [79].

**Figure 53 polymers-13-04393-f053:**
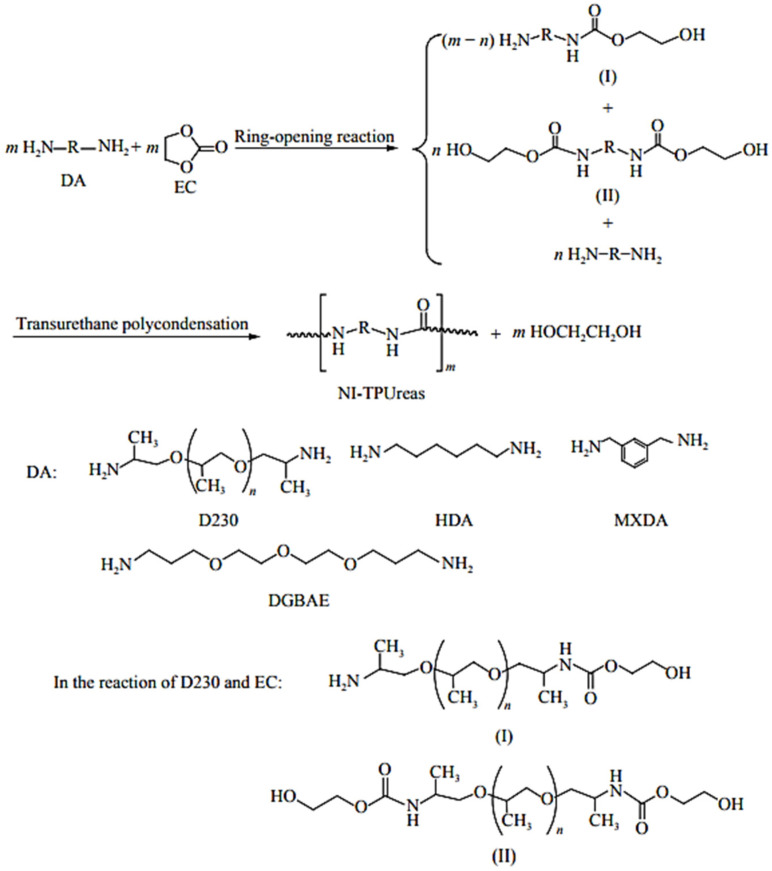
Synthesis of NI-TPUreas from Das and EC. From reference [80].

**Figure 54 polymers-13-04393-f054:**
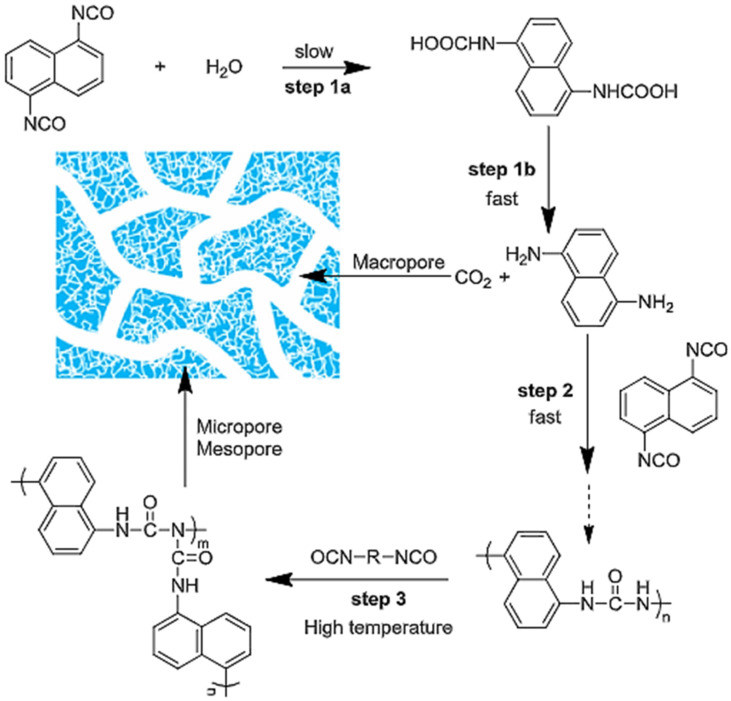
Schematic illustration of formation process for HPUs. From reference [87].

**Table 1 polymers-13-04393-t001:** Monomer compositions and the intrinsic viscosity of the polyurea samples. From reference [42].

Sample	Monomer				
	MDI (g) (M1)	m-PDA (g) (M2)	1,6-HDA (g) (M3)	Mole Ratio of M1:M2:M3	[η] (dL/g)
a	5.0	2.16	0.00	2:2:0	0.62
b	5.0	1.62	0.58	2:1.5:0.5	0.63
c	5.0	1.08	1.16	2:1.0:1.0	0.90
d	5.0	0.54	1.74	2:0.5:1.5	0.33
e	5.0	0.00	2.32	2:0:2	0.23

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
