# Peer review of "Polyureas Versatile Polymers for New Academic and Technological Applications"

_polymers, 2021, doi:10.3390/polym13244393_

Round 1

Reviewer 1 Report

Authors need to be considered the following points before taking a final decision.

  1. The whole manuscript must be revised by a professional English native speaker. Besides the whole manuscript need to rewrite to express their review properly.
  2. In abstract, please don’t use ‘revision/revised’. Also mention a short statement about your findings which can help people working in this area.
  3. Figure 4 mentioned 2 times.
  4. Figure 28 repeated
  5. Make a Table to summarize the synthesis of polyureas, it will help to understand the different techniques.
  6. Make a Table and mention the catalyst, their advantage/disadvantage.
  7. Also give your opinion the ‘’future prospects and challenges ‘’ based on the reviewed articles.

Author Response

Thanks for your comments. Please see the reply in the attachment

Reviewer 2 Report

  1. Abstract of unsufficient quality. It is ill-conceived. It does not encourage reading the work. Moreover, copy-past of abstract is visible. Authors should avoid errors like that while submitting to the journal.
  2. Introduction: International abbreviation of polyurethanes is either PUR either PU. Authors should decide which abbreviation to use and apply it consistently throughout the work.
  3. Polyureas abbreviation has to be differentiated than the one related to polyurethanes. 
  4. Conclusions are poor, should be much improved.
  5. References are not very up to date. Most of them are comming from years below 2017. As the most up to date articles should be considered te ones from up to 5 years back starting from 2021. In my opinion it is obligatry to fulfill this article into up to date articles presenting improvement of state-of-art in polyureas field.

Apart from mentioned comments article presents actual subject of polyureas synthesis and application, thus it should be considered for publication in Polymers journal after performing this revisions.

Author Response

(The authors gave the same response as above.)

Round 2

Reviewer 1 Report

Review is not only the summary of previous works, but also provide suggestion, and future aspects of the reviewed work. They must highlight the limitation of published work, and possible solution of these limitation.

This article is not considerable at the present format. A native speaker must be  revised  this manuscript.  

Author Response

Questions raised by the reviewers are marked through the revised text.

  • Title misspelling corrected Correct the miss spelling of techonological first page
  • Abstract changed the abbreviation for polyureas by PUR and polyurethanes by PU
  • Abstract rephrased, Changed the word revision by review
  • Changed by iminophosphoranes that page 12
  • Question asking for a table was answered by catalyst : the table and the overall

content is given through the text!! Page 13

  • Question requesting the new numbering of figures and tables was answered by tables and figures were numbered in the order they appear through the text!!page 14
  • Pages 18 and 19 reference 89 were included
  • Page 37. The question asking for re-numbering is answered again in the figure caption.
  • Page 57 the question asking for catalysts table was answered by These issues as catalysts syntheses, strategies’ and so are given through the text!
  • Page 66 and 67 The conclusion section as rephrased as marked in the text.

Round 3

Reviewer 1 Report

The manuscript revised properly. Please consider the following points

  1. Some sentences are too long, example- ''Although, Nuclear Magnetic Resonance (NMR) is an important tool to study polymer chemical structure, their use for study PURs is difficult due to their low solubility in most common solvents due to the presence of hydrogen bonds between their chains.'' please revise carefully the whole manuscript.
  2. Please mark the values in 1H NMR IN Figure 19.
  3. Figure 37, 49 are Table!

Author Response

Long phrases were exclude from the text.

Figure 37 and 49 are tables but they were included as figures copied from its relative references. 

Labels in figure 19 were included. 

This manuscript is a resubmission of an earlier submission. The following is a list of the peer review reports and author responses from that submission.

Round 1

Reviewer 1 Report

The authors submitted a review paper to capture the state of the art in Polyureas, which is the reviewer’s perspective is timely and important. Polyureas have gained significant research and industrial attention since the early 2000s despite being used in several other applications (e.g., automotive) before that date. The paper needs a substantial amount of work to be presentable and adequate for consideration. This manuscript must be rejected since it does not meet the minimum standards of academic publishing neither the rigorous standard of J. Polymers. To be clear, the idea of a review paper is timely and great, but the effort to produce a good paper needs to match the soundness of the idea. Below see, some comments, but due to the abrupt transitions from one short paragraph to another, numerous grammatical and punctuation errors, and insufficient citations on each topic, the reviewer could not proceed any further.

In the list of symbols, please be consistent by capitalizing or not in each word contribution to the abbreviation

The abstract can be developed more to highlight the aspects of the review. Also, please consider replacing the word “revision” by the word “review”

The authors stated: “Polyureas have applications in construction field, new materials and several others from health to technical ones. [1]” Please revise this sentence. Polyureas are used in several industrial domains, including the construction field.

The authors stated: “……namely the high durability and strong resistance to atmospheric, chemical, and biological factors. [1]…” Please add other references about the effect the factors you listed on the performance of polyurea. There are a few papers in the past 10 years about the effect of ultraviolet radiation, for example, on the performance of polyurea (as recent as 2020 and 2021).

The logic of selecting some references while neglecting contemporary ones is not clear. For example, Ramirez et al. have several reports about polyurea foam. Why only include one? Are they the only one worked on polyurea foams? Certainly not!

Can the authors define porous polymers? Why they opted for this phrase instead of the more globally accepted foams or cellular solids?

Amirkhizi et al. were not and are not the only group published on the impact mitigation properties of polyurea. There are at least 8 groups around the United States alone that worked extensively on this material. Besides Amirkhizi et al. (ref #12) was only focused on the constitutive modeling. Why only this paper while his group is still working on the topic?

Reviewer 2 Report

Well written review manuscript. The following points should consider before publication:

  1. The manuscript is not match with the title. 'Academic' in title should not reflect in manuscript. Its also not giving clear meaning. This should be removed.
  2. The abstract must be revised with information about the whole manuscript. At least one sentence should state the clear purpose of this review. Also mention very shortly about urea/polyurea.
  3. In title and abstract the author mention about the polyurea application. Please write separately about the application, make a table highlighting the application of polyureas.
  4. Review article is not only summary of articles, but also need to mention the challenges and their possible scientific solution. Please also make a separate section highlighting polyureas challenges and their future aspects.